# EXPLORING EXPERT FAILURES
# IMPROVES LLM AGENT TUNING

## ABSTRACT

Large Language Models (LLMs) have tremendous potential as agents, excelling in tasks requiring multiple rounds of decision-making. For large-scale deployment, a smaller LLM is commonly fine-tuned by learning from teacher-model trajectories and subsequently improving itself via interaction with the environment. A key challenge is that many complex training tasks never yield a successful trajectory (zero reward): the teacher's trajectories fail to solve them, and the student's limited exploration cannot discover one despite many attempts. Without reward signals during training, the student is unlikely to solve similarly difficult test tasks. Applying Rejection Sampling Fine-Tuning (RFT) to WebShop highlights the issue: GPT-4 (the teacher) may succeed on only 36% of the training tasks, and RFT inherently favors actions drawn from those successes. As a result, the student cannot complete most complex tasks for which the teacher does not provide a direct solution because these tasks require more advanced action sequences. To discover reward signals in these complex tasks, we examined the failed teacher trajectories on these challenging tasks, and found that teacher's trajectories often contain valuable guidance—such as plans and key actions—that student seldom used during its exploration. Motivated by this insight, we introduce Exploring Expert Failures (EEF), which uses expert actions to improve the exploration during training and carefully incorporates them into the training by masking out potentially harmful actions to prevent contamination of the learning process. This further allows us to let our student model utilize additional weaker yet more cost-efficient teachers, such as GPT-3.5 Turbo, without inheriting the weaker teacher's suboptimal behaviors. Consequently, EEF successfully resolves many previously unsolvable tasks and significantly enhances agent performance on test tasks. Notably, our approach achieved a remarkable 62% win rate in WebShop, surpassing both RFT (53.6%) and GPT-4 (35.6%). To the best of our knowledge, this establishes a new state-of-the-art, achieving a score of 0.81 on WebShop and 81/100 on SciWorld, two widely used and challenging tasks for evaluating LLM agents.

## 1 INTRODUCTION

Large Language Models (LLMs) (Achiam et al., 2023; Team et al., 2023; Zheng et al., 2023; Xi et al., 2025) have shown immense potential as autonomous agents, capable of excelling in tasks that require multiple rounds of decision-making (Zhou et al., 2023; Yao et al., 2022). To scale these capabilities to practical deployment, researchers often distill the knowledge of large experts (e.g., GPT-4 (Achiam et al., 2023)) into smaller, more efficient models. A common training paradigm is to first imitate expert trajectories and then let the student model improve itself via exploration.

However, a critical challenge arises when many complex training tasks yield no successful trajectories (zero reward): even strong experts such as GPT-4 fail without additional training, and smaller student models, though fine-tuned on positive trajectories of other tasks, still cannot reach reward signals despite multiple attempts, due to their limited exploration capacity. Without reward signals during training, approaches such as reinforcement learning (RL) and rejection-based fine-tuning struggle to prepare student models for similarly challenging test tasks. Take the application of Rejection Sampling Fine-Tuning (RFT) (Touvron et al., 2023; Yuan et al., 2023; Xi et al., 2024) in WebShop (Yao et al., 2022) as an example. In WebShop, GPT-4 succeeds on only 36% of training tasks, and the student is trained solely on successful expert and student trajectories. Because

RFT relies on this limited set of successes, the student struggles with complex tasks that require action sequences beyond those successful demonstrations. This issue of diminishing returns on harder subtasks has also been observed by other researchers (Yuan et al., 2023; Singh et al., 2023), underscoring a key obstacle in training LLM agents.

To tackle this challenge, we examine failed expert trajectories from demanding training tasks. We observe that the initial (prefix) segments often contain valuable plans and actions that the student model seldom generates during exploration. However, since these trajectories ultimately fail, prior works like ETO (Song et al., 2024b) and NAT (Wang et al., 2024a) tend to view their actions as uniformly negative. As a result, rejection-based methods discard these trajectories entirely, while RL approaches often penalize their actions, missing out on these useful cues. Leveraging this insight, we aim to pinpoint and utilize these informative failure segments to broaden the student's reachable states and uncover reward signals in these difficult tasks.

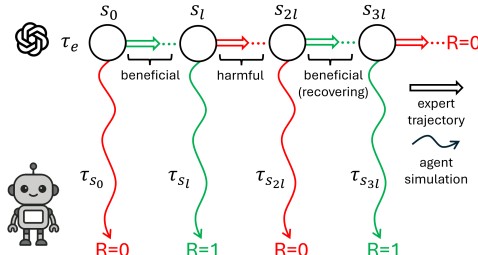

Figure 1: $\tau_e = [s_0, s_1, \dots]$ is a failed expert (GPT-4) trajectory for a challenging task $s_0$ with $R(\tau_e) = 0$. Despite its overall failure, $\tau_e$ contains partial insights to train agent $\pi_\theta$. To identify beneficial actions in $\tau_e$, our method (EEF) performs simulations from expert states $[s_0, s_l, \dots]$ at intervals of length $l$, resulting in $[\tau_{s_0}, \tau_{s_l}, \dots]$, revealing that certain expert actions $a_{0:l-1}$ and $a_{2l:3l-1}$ enable the agent to transit from failed states $s_0, s_{2l}$ to positively-performing states $s_l, s_{3l}$. EEF thus considers these actions (green arrows) beneficial for solving $s_0, s_{2l}$ and adds them into the SFT training dataset while masking the remaining actions (red arrows) which may be potentially harmful.

Guided by these findings, we present Exploring Expert Failures (EEF)—a method that expands exploration by reusing expert actions, acknowledging that failed trajectories frequently encode useful signals. As illustrated in Fig. 1, consider a challenging task where the agent cannot succeed from the initial state $s_0$ despite repeated attempts. By replaying part of the expert's trajectory ($a_{0:l-1}$), the agent can achieve success from the intermediate state $s_l$, thus obtaining a reward signal. Moreover, simulating from expert states allows us to pinpoint beneficial actions (green arrows) while filtering out potentially harmful ones (red arrows). EEF is also designed to mitigate forgetting and simplicity bias by retaining all positive trajectories while employing a selection mechanism to prevent any single task from being over-represented. This design enables EEF to effectively exploit the trajectories of weaker but far more cost-efficient experts such as GPT-3.5 Turbo—30× cheaper than GPT-4—without being significantly hindered by their suboptimal behaviors.

We evaluated EEF in WebShop and ScienceWorld, two highly challenging environments where even strong experts frequently fail. Our results show that effectively harnessing negative expert demonstrations allows EEF to solve subtasks previously unreachable by RFT, while preserving its simplicity. Notably, our method is the first to surpass 0.81 in WebShop and 81 in ScienceWorld, establishing a new state-of-the-art performance.

## 2 BACKGROUND

**Environment for LLM agents.** The environment that includes multiple tasks can be considered as a Markov Decision Process (MDP) with contexts (Hallak et al., 2015): $M = \langle \mathcal{S}, \mathcal{A}, T, R, \mathcal{C} \rangle$. The state space $\mathcal{S}$ consists of states $s \in \mathcal{S}$, each encoding both observable and unobservable parts of the environment. The action space $\mathcal{A}$ encompasses actions the agent can perform, explicitly including internal reasoning steps (e.g., thoughts (Wei et al., 2022b)). The transition function $T : \mathcal{S} \times \mathcal{A} \times \mathcal{S} \to \mathbb{R}$ specifies the probability of transiting to state $s_{t+1}$ after taking action $a_t$ in state $s_t$. The reward function $R : \mathcal{S} \times \mathcal{A} \to \mathbb{R}$ assigns a numerical reward to each state-action pair. In this paper, we consider a sparse reward structure that intermediate steps yield zero reward. For convenience, give a trajectory $\tau = [s_0, a_0, s_1, a_1, \dots, s_T]$, we let $R(\tau)$ denote the final total reward obtained by a trajectory $\tau$. We define the context space $\mathcal{C} \subset \mathcal{S}$ as a set of initial states. Each initial state $s_0 \in \mathcal{C}$ corresponds to a specific task defined by both an instruction and the current environmental situation. During training, the agent can only explore starting from the training tasks $\mathcal{C}_{\text{train}}$.

**Rejection Sampling Fine-Tuning (RFT).** RFT (Yuan et al., 2023) is a training paradigm that begins with expert demonstrations and then iteratively refines the learned policy through exploration. At the beginning, the agent $\pi_\theta$ is trained exclusively on the positive trajectories $R(\tau_e) = 1$ of an expert dataset $\mathcal{D}_e$ to acquire fundamental skills and strategies. In subsequent iterations, for each task $s_0$, the current policy generates $k$ trajectories, and only the positive ones are added to the training set for further fine-tuning. Owing to its simplicity and demonstrable effectiveness, we adopt RFT as our foundational framework, while focusing on refining the exploration process and improving the selection of successful trajectories for fine-tuning. These refinements are orthogonal to the base algorithm and can also benefit other approaches.

**Webshop.** WebShop is an e-commerce simulation that mirrors real-world shopping behavior using over one million Amazon product listings. It comprises multiple tasks, each with the goal of selecting a product that satisfies a given natural-language instruction. We will later utilize Webshop to illustrate our motivation. **Box 1** provided a successful webshop trajectory. Starting from a human instruction, the agent then search (search[white t-shirt]), navigate (click[Next]), select (click[D], click[median]), and purchase ([buy now]) the correct item. This example shows that Webshop is a valuable environment for studying grounded language understanding and multi-step decision-making, as it provides realistic, interactive tasks that require interpreting language and executing sequential actions.

> **Box 1**: WebShop— shopping across over a million real-world Amazon products.
>
> **Human instruction**: I need a m-size white t-shirt under 40 dollars.
> **AI**: search[white t-shirt]
> **Obs**: [Next]
>     [A] Nike white t-shirt, 90 usd.
>     [B] Addidas white t-shirt, 60 usd.
>     [C] Zara white t-shirt, 70 usd.
> **AI**: All too expensive, click[Next]
> **Obs**: [Next] [Prev]
>     [D] H&M white t-shirt, 30 usd.
>     [E] F21 black t-shirt, 40 usd.
>     [F] Uniqlo white jacket, 35 usd.
> **AI** : Only D is a white t-shirt, click[D]
> **Obs**: [back to search]
>     size: [small] [median] [large]
> **AI**: click[median]
> **Obs**: [buy now] [back to product]
> **AI**: click[buy now]
> **Obs**: finished

One of the key skills to solve Webshop is navigation, which is also crucial for many other Web tasks (Yao et al., 2022). Webshop provides two essential navigation actions: **Next** and **Back**. The **Next** action (i.e., Action:click[Next]) allows the agent to access subsequent pages with additional products (as illustrated in **Box 1**), while the **Back** action (i.e., Action:click[back to search]) enables the agent to discard the currently selected product and restart the search. Learning these skills helps the agent recover from a mistake or explore more candidates before making a decision, especially in challenging tasks that even the expert fails to succeed at.

## 3 METHODOLOGY

### 3.1 MOTIVATION

In this subsection, we illustrate the importance of negative expert trajectories in solving hard tasks through an experiment conducted on WebShop, thereby elucidating the motivations behind our proposed approach. In this experiment, we apply RFT to a small model, LLama3 8B (AI@Meta, 2024), with an expert dataset $D_e$ generated by GPT-4, which provides one trajectory per task. Note that the WebShop task is highly challenging, with GPT-4 only achieving approximately 36% success on the training tasks $\mathcal{C}_{\text{train}}$. Consequently, the remaining 64% of tasks lack expert demonstrations and remain OOD after the first iteration of fine-tuning. Although subsequent iterations of RFT allow the agent to generalize and explore successful trajectories for some previously unsolved tasks, roughly 50% of these tasks remain unsolved and thus persist as OOD after multiple iterations (>4). We regard this as a waste, as these OOD tasks and their expert trajectories are accessible yet remain unutilized by RFT agent during training.

To solve the remaining 50% of unsolved tasks, we analyze the failure trajectories produced by both the RFT agent and the expert. In many of these cases, we observe that the RFT agent does not take the necessary navigation actions such as **Next** and **Back**. Consider the example presented in **Box 1**, when the listed products do not meet the requirements (too expensive), the RFT agent still blindly selects one of the options instead of using the **Next** action to check the next pages for more products. Upon inspecting the expert trajectories—though they also result in failure—we find that the expert

---

**Algorithm 1** Exploring Expert Failures (EEF). See visual illustration in Appendix A.3.

1: **Inputs:** Expert dataset: $D_e$, Initial policy: $\pi_\theta$
2: **Parameters:** Finetune Iteration: $I$, Simulation Num: $M$
3: $D^+ \leftarrow \{\tau_e \in D_e : R(\tau_e) = 1\}$        `// D+ is a pos trajectory repository`
4: Optimize $\theta$ with $D^+$ and $\mathcal{L}_{\text{SFT}}$        `// BC on pos expert trajectories`
5: **for** $i = 1, 2, \ldots, I - 1$ **do**
6:    $D_i \leftarrow \{\tau \sim \pi_\theta(\cdot \mid s_0) : s_0 \in C_{\text{train}}\}$        `// Explore initial states with πθ`
7:    **for** $\tau_e = [s_0, s_1, \ldots] \in D_e$ **do**
8:      $l = \lfloor |\tau_e|/(M + 1) \rfloor$        `// Skip length`
9:      $D_i \leftarrow D_i \cup \{\tau \sim \pi_\theta(\cdot \mid s_{m \times l}) : m \in [1, 2, \ldots, M]\}$        `// Explore states of τe`
10:    $D^+ \leftarrow D^+ \cup \{\tau \in D_i : R(\tau) = 1\}$        `// Add pos trajectories to repository`
11:    $D_{s_0} \leftarrow \{\text{get\_traj}(s_0, D^+) : s_0 \in C_{\text{train}}\}$        `// Get solutions for all s0 if exist`
12:    $S_{\text{r}} \leftarrow \text{need\_recover\_states}(D_i)$        `// Get expert states that πθ start failing`
13:    $D_{\text{r}} \leftarrow \{\text{get\_traj}(s, D^+) : s \in S_{\text{r}}\}$        `// Get recovery solutions if exist`
14:    Optimize $\theta$ with $D_{s_0} \cup D_{\text{r}}$ and $\mathcal{L}_{\text{SFT}}$
15: **return** $\pi_\theta$

---

does attempt to use the **Next** action appropriately, but fails due to other factors. In fact, GPT-4 used **Next** on 16.8% of tasks. However, since the majority of these trajectories ultimately fail, they are not utilized by RFT models for training. This represents a missed opportunity, as we observed that RFT agent's policy could successfully complete certain previously unsolved tasks when initialized from states within these trajectories. This indicates that some of the expert actions of these trajectories genuinely provide valuable guidance.

Moreover, we observed that some negative expert trajectories include valuable **recovery** actions, where the expert attempts to correct mistakes made earlier. Take **Box 2** as an example. Although GPT-3.5 Turbo correctly analyzed item A and item B, it overlooked the description of item C and mistakenly selected it (click[C]). In the next turn (marked in green), the expert noticed the mistake and tried to fix it by performing a **Back** action to restart the search. However, although RFT agent can successfully complete this task from the beginning, it failed to perform this recovery action when starting from the expert's incorrect state.

> **Box 2**: A negative GPT-3.5 trajectory. GPT-3.5 first makes a mistake (red), followed by a recovery action (green).
>
> **Human instruction**: I need a living room wall lamp that is in black, and price lower than 100.00 dollars.
> **AI**: search[black living room wall lamp]
> **Obs**: [Next]
>    [A] Generic Raven Decor Crow Lamp Bird Desk Lamp Creative Bedroom Bedside Wall Sconce lamp, $35.
>    [B] Ayux Living Room Dining Room E27 White Deer Head Wall LightGolden Antlers Wall Sconce Rustic Style, $162.
>    [C] Dolity Crow desk Lamp, Bird Lamps Bedroom Resin Crow Desk Bedside Light Wall Sconce Light - Black Table lamp. $33.8
> **AI**: The first product is not a wall lamp, the second product is not in black, so I should skip them and check the next one, click[C]
> **Obs**: [back to search]
>    [buy now]
> **AI**: The third product is a table lamp, not a wall lamp, so I should skip it and check the next one. click[back to search]
> **Obs**: . . .

In other words, expert actions can guide the agent into unfamiliar states while providing useful recovery clues—the agent does not need to master how to reach these states but can still learn the recovery signals, which may prove beneficial when encountering similar mistakes in the future.

Based on these observations, our goal is to include these beneficial expert actions in the RFT fine-tuning process while excluding the remaining expert actions. For instance, in Fig. 1, we consider the action sequence $[a_0, \ldots, a_{l-1}]$ as beneficial for solving state $s_0$, and similarly, $[a_{2l}, \ldots, a_{3l-1}]$ as beneficial for solving state $s_{2l}$ which requires recovery. In contrast, the intermediate actions $[a_l, \ldots, a_{2l-1}]$ are masked, as the simulation failure at $s_{2l}$ suggests that they may not represent useful behavior for the policy to learn.

## 3.2 EXPLORING EXPERT FAILURE (EEF)

Motivated by the observation, we introduce our method, **Exploring Expert Failure (EEF)** (Algo 1), which leverages our current model to explore the environment from expert states. The results of these explorations guide the selection of trajectories and determine specific segments within these trajectories to be used for further fine-tuning. Similarly to previous works such as RFT (Yuan et al., 2023) and ETO (Song et al., 2024b), our method consists of three main phases: (1) **Behavior Cloning** (line 4), (2) **Exploration** (lines 6–10), and (3) **Reinforcement Fine-tuning** (lines 11–15). First, we employ behavior cloning to enhance the capability of a small model. Next, we iteratively perform exploration and reinforcement training phases to further refine the model. In each iteration, the exploration phase simulates all tasks $s_0 \in C_{\text{train}}$ and the states selected from the expert trajectories $\tau_e \in D_e$ with the current policy $\pi_\theta$, and the reinforcement training phase analyzes these simulated trajectories, subsequently training the model with actions identified as beneficial. In Algo.1, the inputs include an expert dataset $D_e$ and an initial policy $\pi_\theta$. The parameters include the number of reinforcement fine-tuning iterations $I$, and the number of simulations $M$ for each expert trajectory per iteration. When the computational budget is limited, one can reduce $M$ to trade off estimation accuracy for efficiency. We provide the details of Behavior Cloning phase in the Appendix A.2 and remaining phases in the following paragraphs.

**Exploration Phase**    In the exploration phase, the current model interacts with the environment to collect trajectories and rewards from the environment. Our exploration phase consists of two types of exploration. The **first type** is the same as RFT, where the policy $\pi_\theta$ explores all tasks $s_0 \in C_{\text{train}}$, as presented in line 6 of Algo. 1. The **second type** of exploration (Algo. 1, lines 7–9) simulates $M$ expert states for each expert trajectory. Our goal is to identify expert actions that are beneficial to our policy, particularly those actions helpful in solving challenging tasks or recovering from expert mistakes. The parameter $M$ controls the computational budget, as some expert trajectories contain an excessive number of states ($|\tau_e| \gg M$), making it computationally impractical to simulate all states. Therefore, given parameter $M$, EEF selects expert states at equal intervals. Specifically, given an expert trajectory $\tau_e = [s_0, a_0, \dots]$, EEF first computes a skip length $l = \lfloor |\tau_e|/(M+1) \rfloor$ and simulates only the selected expert states $[s_l, s_{2l}, \dots, s_{M \times l}]$. All trajectories generated during iteration $i$ are stored in dataset $D_i$ for subsequent analysis. Furthermore, all positive trajectories are retained in the positive trajectory repository $D^+$ to serve as solutions for future training.

**Reinforcement training phase**    After the exploration phase, we have collected numerous positive trajectories in $D^+$, either generated from scratch or from expert states. However, fine-tuning our agent on all the trajectories in $D^+$ is impractical due to high computational costs and may introduce biases toward tasks with more positive trajectories in $D^+$ (Zelikman et al., 2022). Hence, in this phase, our goal is to determine which trajectories in $D^+$ are beneficial for training and to identify the specific actions within these trajectories on which the agent should be trained. To accomplish this, we adopt a two-step strategy: the **important state selection** step, in which EEF identifies states considered important, and the **solution selection** step, where we select a solution from $D^+$ for each important state, if such solutions exist. The selected solutions are the positive trajectories for SFT training. We detail each step in the following paragraphs.

In the **important state selection** step, EEF selects two types of important states. The first type is the initial states $s_0$ of each task, ensuring that these foundational states remain represented to prevent our agent from forgetting how to solve them. The second type consists of states requiring recovery from harmful expert actions. In EEF, we identify these harmful actions through simulations conducted during the exploration phase. Specifically, if the current policy $\pi_\theta$ succeeds when starting from expert state $s_{i-l}$ but fails when starting from expert state $s_i$, we infer that harmful actions exist within the action sequence $a_{i-l:i-1}$ and designate $s_i$ as a state requiring recovery. To avoid overemphasizing any single expert trajectory, EEF selects only the first expert state requiring recovery from each trajectory as part of the second type of important states. Formally, given an expert trajectory $\tau_e = [s_0, s_1, \dots]$ and $M$ simulated states $[s_l, s_{2l}, \dots, s_{M \times l}]$, alongside corresponding trajectories $[\tau_{s_0}, \tau_{s_l}, \tau_{s_{2l}}, \dots]$ generated by the current policy $\pi_\theta$, we define the state $s_{\text{need\_recover}}$ as:

$$s_{\text{need\_recover}} = s_{i^*}, \quad \text{where} \quad i^* = \underset{i \in [l, 2l, 3l, \dots, M \times l]}{\operatorname{argmin}} \{i \mid R(\tau_{s_{i-l}}) = 1, R(\tau_{s_i}) = 0\}.$$

As shown in the equation, $s_{i^*}$ is the first state where the current policy fails ($R(\tau_{s_{i^*}}) = 0$) after previously succeeding ($R(\tau_{s_{i^*-l}}) = 1$), indicating harmful actions within $a_{i^*-l:i^*-1}$. For instance, in Fig. 1, $i^* = 2l$ because the agent succeeds at $s_l$ but fails at $s_{2l}$. This implies expert actions

$[a_l, \ldots, a_{2l-1}]$ may transition the agent to overly challenging states. Thus, if recovery actions enabling success from $s_{2l}$ exist, we aim for the agent to learn these actions, as they may be generalized to other tasks.

Next is the **solution selection** step, where we choose at most one solution for each identified important state from $D^+$ to avoid over-emphasizing any important state. A solution for a given state $s$ is defined as a positive trajectory containing state $s$. In Algo. 1, we use the function get_traj$(s, D^+)$ to get the solution path from the positive trajectory repository $D^+$. If there is no solution path in $D^+$, the function returns None. If there is only one solution path in $D^+$, the function returns the only positive trajectory. If multiple solution paths exist for an important state, we choose the solution path with fewer expert actions. Training the agent on these solution paths with fewer expert-dependent steps minimizes the necessary gradient adjustments and model updates, preserving learning capacity and allowing the model to focus more on other tasks or states. Consider Fig. 1 as an example: state $s_0$ has two solutions: $\tau_{s_l}$ and $\tau_{s_{3l}}$. In this scenario, we select $\tau_{s_l}$ as the solution, as it involves fewer expert-generated actions. Thus, the agent only needs to learn the expert actions $[a_0, \ldots, a_{l-1}]$ since the subsequent actions after the state $s_l$ of $\tau_{s_l}$ are already known to the agent.

After collecting the solution trajectories for the two categories of important states, stored respectively in $D_{s_0}$ and $D_r$ as detailed in Algo. 1, we fine-tune our agent using on $D_{s_0} \cup D_r$ (Algo. 1, line 14). In particular, **when a trajectory is selected as the solution for a specific state $s_i$, the agent is trained only on the actions $[a_i, a_{i+1}, \ldots]$ that occur after state $s_i$.** That is, we disable loss propagation for earlier actions $[a_0, \ldots, a_{i-1}]$ before $s_i$. This selective training prevents the agent from incorporating potentially problematic actions preceding state $s_i$. For instance, in Figure 1, trajectory $\tau_{s_{3l}}$ provides a solution for state $s_{2l}$. Although actions $a_0, a_1, \ldots, a_{2l-1}$ are present within $\tau_{s_{3l}}$, we exclusively train our agent on actions $a_{2l}, a_{2l+1}, \ldots$, as only these subsequent actions are relevant for solving state $s_{2l}$. By focusing learning solely on actions following important states, our agent reduces the risk of imitating suboptimal expert behaviors.

Table 1: Statistics of datasets. **Total expert #**: # of expert trajectories. **Avg Len**: the average length.

|  | Total Expert # | Positive Expert # | Avg Len |
|---|---|---|---|
| Webshop 11k | 11338 | 4106 | 8.26 |
| Webshop 3k | 2835 | 1045 | 8.24 |
| Sciworld 2k | 2120 | 1489 | 20.2 |

## 4 EXPERIMENTS

### 4.1 EXPERIMENTAL SETTINGS

**Datasets** We conducted our experiments in three datasets: WebShop 11k, WebShop 3k, and SciWorld 2k (Table 1). Specifically, WebShop 11k (Yao et al., 2022; Ma et al., 2024) examines our method's effectiveness with extensive data, while WebShop 3k, a random subset of WebShop 11k, evaluates generalization with limited data. Additionally, SciWorld (Wang et al., 2022) tests performance on longer trajectories. For reward metrics, WebShop provides higher rewards when the agent purchases items that satisfy more requirements, whereas SciWorld awards higher rewards based on the completion of predefined subgoals (Appendix A.4). For expert demonstrations, we used GPT-4 0613 to generate trajectories based on human instruc-

Table 2: Methods requiring fine-tuning. **Use Neg**: negative GPT-4 trajectories are used. **FT Iter**: total iter # $I$. **Sim. #**: # of sims per task. **Use GPT-3**: GPT-3.5-Turbo's trajectories ($30\times$ cheaper than GPT-4) are used.

|  | Use Neg | FT Iter | Sim # | Use GPT-3 |
|---|---|---|---|---|
| SFT ALL | ✓ | 1 | 0 | × |
| SFT POS | × | 1 | 0 | × |
| NAT (Wang et al., 2024a) | ✓ | 1 | 0 | × |
| ETO (Song et al., 2024b) | × | 3 | 1 | × |
| RFT (Yuan et al., 2023) | × | 3 | 1 | × |
| RFT $\times 6$ | × | 3 | 6 | × |
| EEF GPT-4 | ✓ | 3 | 6 | × |
| EEF GPT-3&4 | ✓ | 3 | 11 | ✓ |

tions within the WebShop datasets (Liu et al., 2024). Furthermore, in our method EEF GPT-3&4, GPT-3.5-Turbo-generated demonstrations (30 times cheaper than GPT-4) were added to assess our methods under conditions involving weaker expert mixed in the dataset. In contrast, for SciWorld, the demonstrations were sourced directly from AgentGym (Xi et al., 2024). Hence, there is no EEF GPT-3&4 version in SciWorld.

Table 3: Winrates and rewards on three agentic environments. Reward ranges: [0, 1] for WebShop-11k/3k, and [0, 100] for ScienceWorld-2k. Winrate: ratio of achieving maximum reward.

| | FT Iter | Webshop 11k | | Webshop 3k | | ScienceWorld 2k | |
|---|---|---|---|---|---|---|---|
| | | Winrate | Reward | Winrate | Reward | Winrate | Reward |
| GPT-3.5 Turbo | 0 | 23.2% | 0.60 | 23.2% | 0.60 | – | 7.64 |
| GPT-4 | 0 | 35.6% | 0.66 | 35.6% | 0.66 | – | 14.4 |
| SFT ALL | 1 | 37.2% | 0.66 | 39.6% | 0.68 | 53.0% | 68.1 |
| SFT POS | 1 | 46.4% | 0.75 | 39.6% | 0.67 | 61.0% | 76.8 |
| NAT | 1 | 37.2% | 0.66 | 40.4% | 0.68 | 54.0% | 69.2 |
| ETO | 3 | 42.0% | 0.68 | 37.8% | 0.67 | 57.5% | 75.0 |
| RFT | 3 | 52.0% | 0.75 | 38.8% | 0.66 | 61.5% | 74.6 |
| RFT x 6 | 3 | 53.6% | 0.76 | 41.4% | 0.68 | 62.5% | 73.4 |
| EEF GPT-4 | 3 | 58.4% | 0.78 | 46.8% | 0.72 | **68.5%** | **81.3** |
| EEF GPT-3 & 4 | 3 | **62.0%** | **0.81** | **50.0%** | **0.73** | – | – |

**Our method and baselines** We utilize LLAMA3 8B Instruct (AI@Meta, 2024) as the initial model for fine-tuning. Each fine-tuning iteration consists of six epochs. The iteration number is three and we select the best model from all iterations. For our method, we explored five expert states per trajectory ($M = 5$). Our baselines can be categorized into three types. The first type includes models **without fine-tuning**, specifically GPT-3 Turbo and GPT-4. The second type comprises models that are **finetuned without exploration**; within this group, we have three baselines. The "SFT All" baseline learns from all demonstrations, regardless of their correctness. The "SFT POS" baseline learns exclusively from positive demonstrations. The "NAT" (Wang et al., 2024a) baseline trains on the entire dataset, explicitly marking negative trajectories by including an incorrect label in their prompts. The third category of baselines are **finetuned with exploration**. For this category, we selected RFT and ETO (Song et al., 2024b). We tested two RFT variants: "RFT," which conducts a single exploration per task per iteration, and "RFT×6," which explores each task six times with different temperature settings ranging from 0.2 to 0.95. We implement both RFT and RFT×6 using DART-Uniform Tong et al. (2024). In contrast to RFT's use of SFT, ETO employs DPO (Rafailov et al., 2023), where agent-generated trajectories are compared directly with expert demonstrations based on environment reward (still provided) to produce labeled training pairs. PPO is excluded due to its significantly lower performance in preliminary experiments (Song et al., 2024b) compared to ETO.

### 4.2 MAIN RESULTS

Table 3 presents the performance comparison between our proposed EEF method and other baselines in three datasets. The results show a clear performance advantage of our proposed method compared to other baselines, including the GPT-4 baseline, across all evaluated datasets. Specifically, the EEF GPT-4 variant achieves significant improvements over GPT-4, increasing the win rate from 35.6% to 58.4% on Webshop 11k, from 35.6% to 46.8% on Webshop 3k, and notably from lower than 14.4% to 68.5% on SciWorld 2k. Furthermore, compared to the best-performing fine-tuning baseline (RFT × 6), EEF GPT-4 demonstrates additional improvements, raising the win rate from 53.6% to 58.4% on Webshop 11k, from 41.4% to 46.8% on Webshop 3k, and 62.5% to 68.5% on SciWorld 2k. Additionally, when incorporating data from GPT-3 Turbo into EEF (EEF GPT-3 & 4), we observe even greater performance gains, particularly on the Webshop datasets, where the win rate further rises to 62.0% and 50.0% for Webshop 11k and Webshop 3k, respectively. This highlights that our method effectively leverages additional demonstrations from a weaker expert (GPT-3.5 Turbo), achieving notable performance improvements with only a slight increase in cost.

### 4.3 ABLATION STUDIES

**Navigation skills** In this Webshop 11k experiment, we assess whether simplicity bias is mitigated by measuring the percentage of successful tasks involving navigation skills for each method. Specifically, we focus on two types of navigation actions: "Next page" (Next) and "Back to search" (Back),

as shown in Fig. 2. Our methods, EEF GPT-4 and EEF GPT-3&4, exhibit superior proficiency in utilizing these navigation skills compared to baseline models. In addition, we investigated the attempt rate of using navigation actions to determine whether EEF actually learns navigation skills from the failure trajectories of GPT3&4. The results are shown in Table 4 of Appendix A.6. In particular, while GPT-3.5 and GPT-4 display a lower percentage of successful tasks that involve navigation actions, they actually show relatively high attempt rates. For example, GPT-4 attempted to use the Next action in 16.8% of the tasks. This indicates that GPT-4 recognizes the potential usefulness of such navigation actions but lacks the ability to execute them effectively. In contrast, our method not only identify these beneficial actions in the negative expert trajectories but also convert them into successful task completions through more effective skill utilization.

**Case studies**    To substantiate the necessity of navigation skills for challenging tasks, we examine specific tasks where **(1)** our method succeeded by effectively employing these skills, **(2)** RFT methods failed due to the absence of these skills, and **(3)** GPT-4 applied these skills but ultimately failed. Some representative samples of these tasks, along with a side-by-side comparison of trajectories generated by the RFT method and our method, can be found in Appendix A.10 and A.11. Upon examining the trajectores of these type tasks, it becomes evident that navigation skills are crucial. For tasks that require the Next action, the products initially presented do not satisfy user requirements. For instance, listed products may have incorrect attributes such as an inappropriate price range

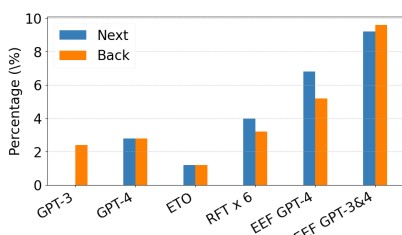

Figure 2: The percentage of successful tasks that involve the use of navigation skills: **Next** and **Back**.

or incorrect colors (e.g., ask for petal green but get petal blue). Similarly, for tasks that require the Back action, the items initially selected appear suitable on the search page, but ultimately lack specific attributes such as desired sizes or colors upon closer inspection on the product page. (e.g., ask for a 250ml shampoo but only find 500ml or 200ml options). Another notably challenging task involved a user's request for a high-power sound system. Here, the agent needed to recognize that a portable 16W soundbar was insufficient and consequently navigate to the next page to identify a more suitable product. These case studies clearly demonstrate that our approach significantly mitigates simplicity bias and is able to employ advanced navigation skills.

**Efficiency Analysis**    In this section, we evaluate the exploration efficiency of our proposed method by varying the number of simulations, specifically setting $M = 1, 2, 5$. We compared these results with a baseline exploration method that utilizes different temperatures within the Webshop 3k environment. To ensure fairness in our comparison, we employed the same model (SFT POS) for conducting explorations, and subsequently utilized the resulting data to finetune a new model with only one iteration. The results presented in Fig. 3 clearly demonstrate the superior performance of our method relative to the baseline exploration approach. Specifically, the baseline method fails to achieve a 40% winrate even when employing up to 20 simulations. In contrast, our method achieves a 40% winrate using merely 2 simulations (when $M = 1$) and consistently improves performance with increased simulation budgets. Furthermore, our findings indicate that exploring the trajectories generated by GPT-3 consistently yields better results than exploring those generated by GPT-4. This indicates that the GPT-3.5 Turbo trajectories may have more things to learn for the model trained only by GPT-4.

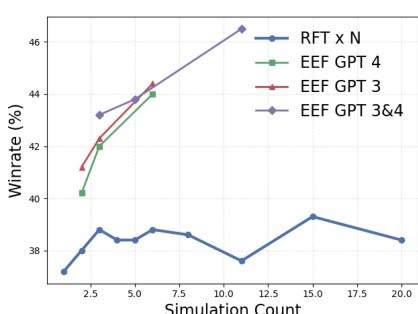

Figure 3: Win rates of different methods training the same base model for one iteration under varying simulation budgets (exploration cost) by adjusting $N$ or $M$. **RFT × N**: Initial state $s_0$ simulated $N$ times. **EEF**: Different expert state simulation numbers $M = 1, 2, 5$ on different expert datasets.

**Different Models** To further validate the effectiveness and generalization of our method, we conducted additional experiments using one of the latest models, mistral-7b-v0.3, on the Webshop 3k dataset. The results demonstrate that our approach consistently outperforms existing baselines. Specifically, our method achieved win rates of 46.8% and 48.8% with the "GPT-4" and "GPT-3&4" models, respectively, compared to only 42.2% obtained by RFTx6. These findings underscore the robust generalization capability of our approach across different initial LLMs.

## 5 RELATED WORK

Recent advances in large language models (LLMs) have prompted researchers to build LLM-based agents for multi-step tasks, exploiting the models' emergent abilities (Achiam et al., 2023; Wei et al., 2022a). Broadly, existing approaches can be categorized into methods without fine-tuning (zero-shot or few-shot), methods with fine-tuning but no exploration, and methods with exploration-based fine-tuning. Zero-shot and few-shot methods typically focus on prompt engineering to reduce errors and hallucinations (Huang et al., 2022a;b; Yao et al., 2023; Wang et al., 2023a). Approaches that use fine-tuning without exploration often rely on filtering suboptimal expert trajectories based on rewards (Zeng et al., 2023; Chen et al., 2023; 2024), or adding a negative label on the prompt of negative demonstrations (Wang et al., 2024a). and some also train on a large set of tasks to learn general concepts transferable to tasks with limited demonstrations (Zhang et al., 2024; Song et al., 2024a). Exploration-based fine-tuning adds newly discovered positive samples to the supervised fine-tuning dataset (Aksitov et al., 2023; Xi et al., 2024); experiments have shown that training on self-generated data can be more sample-efficient (Setlur et al., 2024). Other variants (e.g., ETO) use DPO to reduce the likelihood of generating negative trajectories (Song et al., 2024b). Note that ETO, as well as many methods (Xiong et al., 2024) that rely on ETO's dataset, use only high-reward expert trajectories. However, many of these approaches treat the entire trajectory with a single final reward, which can be simplistic.

More recent methods adopt a stepwise analysis of trajectories for better finetuing (Wang et al., 2023b; 2024b; Ma et al., 2023; Havrilla et al., 2024; Xiong et al., 2024). For example, by training a model to identify critical steps or highly rewarded and then focus training only on those steps (Chen et al., 2025; Wang et al., 2025). Another strategy assumes that the expert is always correct, applying DPO at each step (Deng et al., 2024); but this is not viable when the expert frequently fails. Stepwise DPO (Lai et al., 2024) assumes that if a state contains both positive and negative trajectories, the agent should learn from the positive action. Yet implementing stepwise DPO on the negative expert trajectories still differs from our approach. Specifically, given a state $s_i$ and two trajectories $\tau_{s_i}^{+}$ and $\tau_{s_i}^{-}$ labeled as chosen and rejected, stepwise DPO increases the probability of actions after $s_i$ in $\tau_{s_i}^{+}$, while decreasing it for $\tau_{s_i}^{-}$. In contrast, our method emphasizes beneficial actions *before* $s_i$ that contribute to the successful simulation $\tau_{s_i}^{+}$.

## 6 CONCLUSION

We present EEF as a practical and effective exploration method that increases the likelihood of encountering reward signals in OOD subtasks by converting signals from failed expert trajectories into training guidance. For subtasks that exist positive solutions, EEF leverages harmful expert actions to drive exploration into unfamiliar states, and we show that valuable recovery cues can be distilled from subsequent steps. After collecting a more diverse pool of trajectories, EEF selects a compact subset of positives while masking risky segments to form supervised data. The approach learns effectively from strong experts (e.g., GPT-4) and can also benefit from weaker experts such as GPT-3.5. Across WebShop and ScienceWorld, EEF attains state-of-the-art results while preserving the simplicity of RFT—training solely with an SFT loss and no reward model—improving practicality and reducing hyperparameter tuning. Future work spans three directions. First, EEF-identified positive trajectories can provide explicit reward signals for RL, helping prevent collapse when rewards are sparse. Second, since EEF can be interpreted as a form of tree search over expert trajectories, it can be extended with more advanced search strategies (e.g., MCTS) to improve the chance of reaching rewarding states. Third, it is important to study how to allocate budget between high-quality and low-cost experts, as both produce informative trajectories but entail different costs. Together, these directions position EEF as a promising stepping stone toward more powerful exploration frameworks in the future.

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

# A APPENDIX

## A.1 PROBLEM DEFINITION

We restate our problem setting as follows:

- **Efficiency goal:** We aim to fine-tune a lightweight model to solve practical, multi-subtask problems such as WebShop quickly.
- **Expert trajectories:** Due to high human annotation costs, we generate only one GPT-4 trajectory per training subtask. During training, the agent cannot query GPT-4 because of its high cost.
- **Continual OOD tasks:** Tasks of WebShop are highly challenging. Many tasks fail not only in the GPT-4 trajectory but also after extensive exploration, never reaching any reward signal. Consequently, these tasks remain absent from training data and are considered continually out-of-distribution (OOD).
- **Limitation of reward-based methods:** If repeated standard exploration fails to reach any reward signal, reward-based methods (e.g., RL) face fatal limitations. This motivates the use of surrogate rewards (e.g., curiosity or progress) in prior works to boost exploration efficiency.
- **Exclusion of reasoning-heavy methods:** We temporarily exclude reasoning methods (e.g., GRPO), since fine-tuning lightweight models inherently emphasizes efficiency. Current reasoning approaches usually require more than a hundred iterations with lengthy thought processes, which directly conflicts with our efficiency constraints.

## A.2 BEHAVIOR CLONING PHASE

In this phase, we train our policy to acquire fundamental skills by imitating expert behaviors. First, like RFT, we reject the negative and select only positive expert trajectories as our training dataset $D^+$ in Algo 1 line 3. Next, we fine-tune our LLM model $\pi_\theta$ with auto-regressive loss in Algo 1 line 4. Specifically, given an expert trajectory $\tau_e = (s_0, a_0, s_1, \dots)$, we first convert it into a pure text sequence $\tau = (o_0, a_0, o_1, \dots)$, that only includes observations and actions. Next, we concatenate the observations $o$ and actions $a$ as a string and convert them to a sequence of tokens $\mathbf{t} = [t_0, t_1, \dots, t_L]$, where $L$ is the length of the sequence. Since $\pi_\theta$ should only train on the action part of the sequence, for each token $t_l$, we define the loss mask as $m_l = \mathbb{1}(\exists i \quad \text{s.t.} \quad t_l \in a_i)$, where the indicator $\mathbb{1}$ returns 1 if $t_l$ belongs to one of the actions $a_i$. Finally, the masked autoregressive loss is defined as:

$$\mathcal{L}_{\text{SFT}}(\pi_\theta) = -\sum_l m_l \times \log \pi_\theta(t_l | t_{<l}).$$

With $\mathcal{L}_{\text{SFT}}$, we update the weights $\theta$ of our policy $\pi_\theta$ in Algo. 1 line 4 to obtain a decent policy to conduct the following exploration.

## A.3 EEF ILLUSTRATION

**Exploration Phase:** Agent explores environment, generating trajectories. EEF stores all positive (successful) ones in a repository to prevent forgetting solutions in future iterations.

**Reinforcement Training Phase:**

- **Important State Selection:** Identify initial state $s_0$ and the first failed "expert state." $s_{2l}$
- **Repository Query:** Search positive trajectories for solutions to these states.
- **Solution Handling:** If found, select the one closest to current policy; mask actions before the important state to avoid harmful learning.

## A.4 DATASETS

**Webshop** WebShop (Yao et al., 2022) is a simulated online shopping environment containing 1.18M real-world Amazon products, with shopping tasks provided as human instructions. Each in-

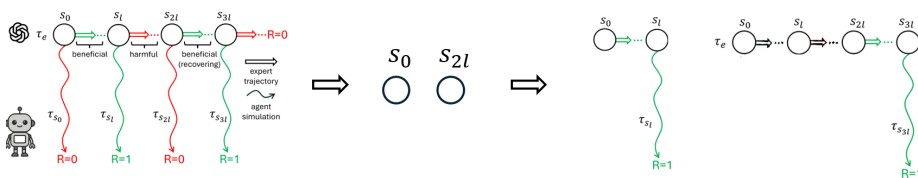

Figure 4: Illustration of the training pipeline. From left to right: (1) Exploration, (2) Selection of important states, (3) Collection of positive trajectories for these states with potentially harmful actions masked.

struction specifies a ground-truth product and detailed attribute requirements. We utilize the Agentboard Ma et al. (2024) version of WebShop.

The reward system is based on four criteria: 1) coarse-grained product category match, 2) fine-grained category match, 3) product title match, and 4) price match. However, we observed some inconsistencies in the scoring system. Specifically, products with multiple matching attributes receive higher scores, but the system equally weights all attributes without considering their varying importance.

For instruction (task) separation, we follow Agentboard (Ma et al., 2024) and Agentlite (Liu et al., 2024), using the same 251 human instructions as testing instructions (stored in test.jsonl). We additionally partitioned 750 human instructions (IDs: 2250–3000) as our validation set and used the remaining instructions for training. The Webshop11k dataset includes all training instructions, while Webshop3k only includes instructions whose IDs are divisible by four.

Both GPT4-0613 and GPT3.5-Turbo expert datasets are generated using the Agentlite (Liu et al., 2024) framework, as it reports one of the highest GPT-4 test results (0.676). For each task $s_0$, we perform one simulation run with each of GPT4-0613 and GPT3.5-Turbo.

**SciWorld** ScienceWorld (Wang et al., 2022) is a text-based virtual environment designed for performing elementary science experiments across 10 distinct task types, including thermodynamics and electrical circuits. Agents must observe their surroundings and identify appropriate items to complete each task. Each task comprises multiple optional subgoals, with the overall reward determined by how many of these subgoals are achieved. Additionally, each task features various variations, each treated as a separate task .

Teleport actions, which move agents instantly between rooms, are disallowed to maintain real-world plausibility. Following prior work (Song et al., 2024b; Xi et al., 2024), we exclude Task-9 and Task-10 due to their excessively long solution trajectories.

The rewards are determined by completing defined subgoals. We re-annotated tasks to include critical observations and essential steps, ensuring more accurate progress measurement. Each subgoal, whether sequential or observational, contributes meaningfully to the final score. This approach mitigates issues from uneven subgoal distribution and better reflects the agent's navigation, task comprehension, and common-sense reasoning abilities.

For task division and expert demonstrations, we adopt the separation strategy used by Agent-Gym (Xi et al., 2024). Since existing models cannot reliably achieve high performance using the provided golden paths from the ScienceWorld repository, AgentGym employs GPT-4-Turbo to generate thought processes along golden paths for 22 interaction types of manageable length. Reward is used as the evaluation metric, with the maximum interaction rounds capped at 30.

A.5 OUR METHOD AND BASELINES

The primary distinction between EEF GPT-4 and EEF GPT-3&4 lies in their expert datasets. For EEF GPT-4, the expert dataset contains a single GPT-4 trajectory per training subtask. In contrast, EEF GPT-3&4's includes one GPT-4 trajectory and one GPT-3.5 trajectory per subtask. Addition-

ally, during the supervised fine-tuning (SFT) phase (Algo.1 Line 4), EEF GPT-3&4 fine-tunes exclusively on GPT-4 trajectories, despite the presence of GPT-3.5 trajectories in its dataset. All other aspects are identical to Algorithm 1.

Our RFT use the DART-Uniform setting: every sub-task is sampled with equal probability, regardless of how many positive examples it contains. We also evaluated DART-Prop2Diff, which up-weights harder sub-tasks, but its results were almost identical to DART-Uniform.

Both variants of DART fail because they rely solely on the current policy for exploration, leaving subtasks without positive examples out-of-distribution (OOD). Specifically, DART only increases the sampling probability for subtasks that already have a reward signal, shifting it from low to high. In contrast, EEF addresses subtasks that initially have zero positives by seeding exploration with expert-failed trajectories (first half from expert states, second half from policy roll-outs), thereby transitioning them to positive examples and overcoming simplicity bias.

This decision is based on preliminary experiments, which indicate that including GPT-3's positive trajectories in this phase leads to decreased performance. For hyperparameter of training, we used a batch size of 64 and a learning rate of 5e-5. All experiments were conducted with four NVIDIA A6000 GPUs.

## A.6 NAVIGATION ACTIONS' SUCCESS AND ATTEMPTED RATE

Table 4 illustrates simplicity bias, showing that expert successes use minimal navigation skills, while expert failures involve more navigation actions (e.g., next/back) that can be learned from. The Next/Back Attempt Rate is the proportion of test trajectories with at least one next or back action, and the Next/Back Success Rate is the proportion of test trajectories containing at least one next/back action that end in a successful outcome (Each test task has one trajectory.)

Table 4: Performance comparison in terms of pos and all metrics. Columns 2-3 indicate the proportion of test subtasks successfully solved using Next or Back, respectively. Columns 4-5 show the proportion of attempts made by agents using these navigation skills.

|  | Success (%) | | Attempt (%) | |
| --- | --- | --- | --- | --- |
|  | Next | Back | Next | Back |
| GPT-3.5 Turbo | 0.0 | 2.4 | 3.2 | 16.8 |
| GPT-4 | 2.8 | 2.8 | 16.8 | 14.0 |
| SFT ALL | 2.8 | 1.6 | 16.8 | 9.6 |
| SFT POS | 4.4 | 2.0 | 13.2 | 4.0 |
| ETO | 1.2 | 1.2 | 9.6 | 3.2 |
| RFT | 4.0 | 3.2 | 9.6 | 8.4 |
| RFT × 6 | 4.0 | 3.2 | 9.2 | 7.6 |
| EEF GPT-4 | 6.8 | 5.2 | 13.2 | 12.0 |
| EEF GPT-3&4 | 9.2 | 9.6 | 13.2 | 14.0 |

## A.7 SIMPLICITY BIAS IN SCIWORLD

In the SciWorld scenario, EEF's actions such as "move wind generator", "move to the greenhouse", and "move egg crocodile egg in inventory to purple box" are never utilized by SFT/RFT agents. Additionally, these SFT/RFT frequently attempt tasks prematurely in unsuitable rooms (e.g., selecting a lamp in the bedroom when searching for a plant), whereas EEF agents continue exploring other rooms—similar to using the next action in WebShop.

## A.8 COULD EEF UTILIZE ONLY SELF-GENERATED TRAJECTORIES INSTEAD OF RELYING ON EXPERT TRAJECTORIES?

We tested a self-generated variant using high-temperature sampling of the current policy as expert data. This setup outperforms vanilla RFT. However, high-temperature exploration alone is insufficient, as the model still exhibits navigation gaps. Thus, EEF primarily relies on expert traces. Unlike fine-tuning methods that only learn from successful expert runs, EEF is less sensitive to expert quality, as it can extract valuable segments from both successful and failed expert trajectories. Additionally, expert data can be generated using the previous policy, potentially enhancing diversity in the training data.

## A.9 MORE RELATED WORKS

**Why not reasoning RL method?** Our approach is orthogonal to RL algorithms. Our core issue is that standard exploration methods are insufficient. In many tasks, the reward is not just sparse but strictly zero, making standard exploration ineffective at reaching any reward signals even after multiple simulations. Therefore, we introduced a more effective exploration strategy to discover these signals and proposed a simple but concrete method to leverage successful trajectories, demonstrating that these signals are indeed usable. Once successful trajectories are obtained, RL methods might also be able to utilize those positive trajectories.

In addition, while RL methods have shown success in various LLM-related tasks, they are prone to instability. For example, the ETO paper notes that PPO suffers from inherent instability in RL optimization and sometimes is worse than the SFT method. Even specialized RL methods for LLM

agents, such as ETO (online DPO) and IPR (online stepwise DPO), experience performance degradation after a few iterations, underscoring their instability. In contrast, our method achieves consistent stability, maintaining performance without noticeable declines after reaching its plateau. Our approach benefits from the stability of an SFT loss, yet avoids indiscriminately learning from all demonstrations—including harmful actions, a common pitfall of standard SFT/RFT methods.

## A.10 CASE STUDIES OF NEXT ACTION

### Next Example 1: RFT, reward = 0.66

Task:i am looking for resilient memory foam loveseat sofa

Action:Think["response": "I should first use Search to search for a resilient memory foam loveseat sofa."]

Observation: OK

Action:search["product": "resilient memory foam loveseat sofa"]

Observation: WEB PAGE:
[Back to Search]
Page 1 (Total results: 15)
[Next]
[B08737G8HG] Modern Fabric Double Sofa Couch Loveseat Linen Accent Chair Comfy Upholstered Arm Lounge Chair for Bedroom Living Room $279.99
[B09NS47YJ7] Sectional Couches for Living Room Faux Leather Office Couch Sets Mid Century Modern Small Sofa for Bedroom Mini Settee Loveseat Comfy for Small Apartment(Loveseat, Brown) $512.0
[B09F5J244F] Hydeline Erindale Power Recline, Headrest, Lumbar, Top Grain Leather Sofa Couch with Built in USB Ports, 88", Cognac $2699.0

Action:Think["response": "The first product seems to be the most relevant to the requirements. I should click the first product."]

Observation: OK

Action:click["button": "B08737G8HG"]

Observation: WEB PAGE:
[Back to Search]
[Prev] Modern Fabric Double Sofa Couch Loveseat Linen Accent Chair Comfy Upholstered Arm Lounge Chair for Bedroom Living Room Price: $279.99

Action:click["button": "Buy Now"]

Observation: shopping is finished.

### Next Example 1: EEF, reward = 1.00

Task:i am looking for resilient memory foam loveseat sofa

Action:Think["response": "I should first use Search to search for resilient memory foam loveseat sofa."]

Observation: OK

Action:search["product": "resilient memory foam loveseat sofa"]

Observation: WEB PAGE:
[Back to Search]
Page 1 (Total results: 15)
[Next]
[B08737G8HG] Modern Fabric Double Sofa Couch Loveseat Linen Accent Chair Comfy Upholstered Arm Lounge Chair for Bedroom Living Room $279.99
[B09NS47YJ7] Sectional Couches for Living Room Faux Leather Office Couch Sets Mid Century Modern Small Sofa for Bedroom Mini Settee Loveseat Comfy for Small Apartment(Loveseat, Brown) $512.0
[B09F5J244F] Hydeline Erindale Power Recline, Headrest, Lumbar, Top Grain Leather Sofa Couch with Built in USB Ports, 88", Cognac $2699.0

Action:Think["response": "None of the products listed are resilient memory foam loveseat sofas. I should click Next to see more options."]

Observation: OK

Action:click["button": "Next"]

Observation: WEB PAGE:

[Back to Search]
Page 2 (Total results: 15)
[Prev]
[Next]
[B09NS26T9M] Faux Leather Office Couch Sets Mid Century Modern Sectional Small Sofa for Bedroom Mini Settee Loveseat Comfy for Living Room Small Apartment(3-Seat Sofa, Black) $512.0
[B09QMF83B5] 56-inch Small Loveseat Sofa, Modern Couch with Removable Cushion, Upholstered Living Room Sofa with Rose Golden Metal Legs, Leopard Teal $259.99
[B0872G9J57] Poppy Sofa - Nelson Sectional/Sofa/Loveseat/Loveseat Sectional/Armchair 42 Colors - Resilient Memory Foam - Made in US - Mid Century Modern (Sand Cross Weave Upholstery) SF031720036E1 $1751.0

Action:Think["response": "The third product is a resilient memory foam loveseat sofa. I should click on the third product."]

Observation: OK

Action:click["button": "B0872G9J57"]

Observation: WEB PAGE:
[Back to Search]
[Prev]
size [armchair][loveseat][sectional][sectional loveseat][sofa]
Price: $1751.0
[Buy Now]

Action:click["button": "loveseat"]

Observation: WEB PAGE: You have clicked loveseat.

Action:click["button": "Buy Now"]

Observation: shopping is finished.

## Next Example 2: RFT, reward = 0.66

Task:i need a detangler hair brush that stimulates hair growth. choose the purple one, and price lower than 50.00 dollars

Action:Think["response": "I should first use Search to search for a detangler hair brush that stimulates hair growth."]

Observation: OK

Action:search["product": "detangler hair brush that stimulates hair growth"]

Observation: WEB PAGE:
[Back to Search]
Page 1 (Total results: 15)
[Next]
[B08KYNCDVG] Zolf Natural Bristle Detangling for Wet/Dry Hair, for Women, Men, and Kids. Includes brush cleaner. $19.77
[B07QW12YR9] Hair Brush, Natural Boar Bristle Hair Brush, Wooden Bamboo Hair Brush for Women Mens, Paddle Brush Curly Hair Brush for Thick Hair & Anti Static Detangling Best Paddle Brush for Reducing Hair Breakage. $9.99
[B09189QFJ9] Hair Brush, Natural Boar Bristle Hair Brush, Wooden Bamboo Hair Brush for Women Mens, Paddle Brush Curly Hair Brush for Thick Hair & Anti Static Detangling Best Paddle Brush for Reducing Hair Breakage $9.99

Action:Think["response": "The first product seems to be the most relevant to the requirements. I should click the first product."]

Observation: OK

Action:click["button": "B08KYNCDVG"]

Observation: WEB PAGE:
[Back to Search]
[Prev]
Zolf Natural Bristle Detangling for Wet/Dry Hair, for Women, Men, and Kids. Includes brush cleaner.
Price: $19.77

Action:click["button": "Buy Now"]

## Next Example 2: EEF, reward = 1.0

Task:i need a detangler hair brush that stimulates hair growth. choose the purple one, and price lower than 50.00 dollars

Action:Think["response": "I should first use Search to search for a detangler hair brush that stimulates hair growth."]

Observation: OK

Action:search["product": "detangler hair brush that stimulates hair growth"]

Observation: WEB PAGE:
[Back to Search]
Page 1 (Total results: 15)
[Next]
[B08KYNCDVG] Zolf Natural Bristle Detangling for Wet/Dry Hair, for Women, Men, and Kids. Includes brush cleaner. $19.77
[B07QW12YR9] Hair Brush, Natural Boar Bristle Hair Brush, Wooden Bamboo Hair Brush for Women Mens, Paddle Brush Curly Hair Brush for Thick Hair & Anti Static Detangling Best Paddle Brush for Reducing Hair Breakage. $9.99
[B09189QFJ9] Hair Brush, Natural Boar Bristle Hair Brush, Wooden Bamboo Hair Brush for Women Mens, Paddle Brush Curly Hair Brush for Thick Hair & Anti Static Detangling Best Paddle Brush for Reducing Hair Breakage $9.99

Action:Think["response": "None of the products listed stimulate hair growth or are purple. I should click Next to see more options."]

Observation: OK

Action:click["button": "Next"]

Observation: WEB PAGE:
[Back to Search]
Page 2 (Total results: 15)
[Prev]
[Next]
[B07WN2DJSL] Curved Vented Boar Bristle Styling Hair Brush, Single White Brush Anti-static Detangler, Wet or Dry Use, Fast Blow Drying, Use on Long or Short Hair. (1 Pack, White) $7.99
[B07VQDM2CF] Oleh-Oleh Wet and Dry Brush Pro Detangle Hair Brush + Folding Mirror Mini Pop Up Hairbrush (BLUE+PURPLE). Healthy to scalp. $16.99
[B00S1L6590] Giorgio GION2R Neon Red 6.25 inch Gentle Touch Detangler Hair Brush for Men Women & Kids. Soft Bristles for Sensitive Scalp. Wet and Dry for all Hair Types. Scalp Massager Brush Stimulate Hair Growth $14.95

Action:Think["response": "The second product is purple and stimulates hair growth. I should click on the second product."]

Observation: OK

Action:click["button": "B07VQDM2CF"]

Observation: WEB PAGE:
[Back to Search]
[Prev]
color [blue+purple][blue+red+light blue][purple][purple+blue][rose+blue+purple][set1]
Oleh-Oleh Wet and Dry Brush Pro Detangle Hair Brush + Folding Mirror Mini Pop Up Hairbrush (BLUE+PURPLE)
Price: $16.99
[Buy Now]

Action:click["button": "purple"]

Observation: WEB PAGE: You have clicked purple.

Action:click["button": "Buy Now"]

Observation: shopping is finished.

## A.11 CASE STUDIES OF BACK ACTION

### Back Example 1: RFT, reward = 0.66

Task:i want a red office chair ergonomic gaming chair with lumbar support, and price lower than 140.00 dollars

Action:Think["response": "I should first use Search to search for a red office chair ergonomic gaming chair with lumbar support."]

Observation: OK

Action:search["product": "red office chair ergonomic gaming chair with lumbar support"]

Observation: WEB PAGE:
[Back to Search]
Page 1 (Total results: 15)
[Next]
[B098DG5F8T] Tyyps Ergonomic Gaming Chair -250lb High Back PC Racing Desk Chair Adjustable Reclining Computer Chair with Lumbar Support Armrest Headrest Task Rolling Swivel Chair Game Chair for Adult Teen, Black $52.95
[B08PPGC1C4] Gaming Chair Office Chair Desk Chair Ergonomic High Back Computer Chair with Lumbar Support Flip-up Arms Headrest PU Leather Swivel Task Chair for Women Men Adult, Pink $105.99
[B097BG6P5Q] Executive PU Leather Office Chair, Ergonomic High Back Managerial Chairs with Lumbar Support Armrest Swivel, Adjustable Computer Gaming Chair for Office Home Gaming Room Red $162.99

Action:Think["response": "The third product is a red office chair with lumbar support, but it's over the budget. I should click the first product."]

Observation: OK

Action:click["button": "B098DG5F8T"]

Observation: WEB PAGE:
[Back to Search]
[Prev]
Tyyps Ergonomic Gaming Chair -250lb High Back PC Racing Desk Chair Adjustable Reclining Computer Chair with Lumbar Support Armrest Headrest Task Rolling Swivel Chair Game Chair for Adult Teen, Black
Price: $52.95
[Buy Now]

Action:click["button": "Buy Now"]

Observation: shopping is finished.

## Back Example 1: EEF, reward = 1.0

Task:i want a red office chair ergonomic gaming chair with lumbar support, and price lower than 140.00 dollars

Action:Think["response": "I should first use Search to search for a red office chair ergonomic gaming chair with lumbar support."]

Observation: OK

Action:search["product": "red office chair ergonomic gaming chair with lumbar support"]

Observation: WEB PAGE:
[Back to Search]
Page 1 (Total results: 15)
[Next]
[B098DG5F8T] Tyyps Ergonomic Gaming Chair -250lb High Back PC Racing Desk Chair Adjustable Reclining Computer Chair with Lumbar Support Armrest Headrest Task Rolling Swivel Chair Game Chair for Adult Teen, Black $52.95
[B08PPGC1C4] Gaming Chair Office Chair Desk Chair Ergonomic High Back Computer Chair with Lumbar Support Flip-up Arms Headrest PU Leather Swivel Task Chair for Women Men Adult, Pink $105.99
[B097BG6P5Q] Executive PU Leather Office Chair, Ergonomic High Back Managerial Chairs with Lumbar Support Armrest Swivel, Adjustable Computer Gaming Chair for Office Home Gaming Room Red $162.99

Action:Think["response": "The third product is a red office chair with lumbar support, but it's over the budget. I should click the first product to see if it has a red color option."]

Observation: OK

Action:click["button": "B098DG5F8T"]

Observation: WEB PAGE:
[Back to Search]
[Prev]
Tyyps Ergonomic Gaming Chair -250lb High Back PC Racing Desk Chair Adjustable Reclining Computer Chair with Lumbar Support Armrest Headrest Task Rolling Swivel Chair Game Chair for Adult Teen, Black
Price: $52.95
[Buy Now]

Action:Think["response": "The product does not have a red color option. I should go back to the search results and try the next product."]

Observation: OK

Action:click["button": "Back to Search"]

Observation: WEB PAGE: Let us begin shopping [Search]

Action:search["product": "red office chair ergonomic gaming chair with lumbar support"]

Observation: WEB PAGE:

[Back to Search]
Page 1 (Total results: 15)
[Next]
[B098DG5F8T] Tyyps Ergonomic Gaming Chair -250lb High Back PC Racing Desk Chair Adjustable Reclining Computer Chair with Lumbar Support Armrest Headrest Task Rolling Swivel Chair Game Chair for Adult Teen, Black $52.95
[B08PPGC1C4] Gaming Chair Office Chair Desk Chair Ergonomic High Back Computer Chair with Lumbar Support Flip-up Arms Headrest PU Leather Swivel Task Chair for Women Men Adult, Pink $105.99

[B097BG6P5Q] Executive PU Leather Office Chair, Ergonomic High Back Managerial Chairs with Lumbar Support Armrest Swivel, Adjustable Computer Gaming Chair for Office Home Gaming Room Red $162.99

Action:click["button": "B08PPGC1C4"]

Observation: WEB PAGE:
[Back to Search]
[Prev]
color [blue][camo][grey][pink][red][white] Gaming Chair Office Chair Desk Chair Ergonomic High Back Computer Chair with Lumbar Support Flip-up Arms Headrest PU Leather Swivel Task Chair for Women Men Adult, Pink Price: $105.99 [Buy Now]

Action:click["button": "red"]

Observation: WEB PAGE: You have clicked red.

Action:click["button": "Buy Now"]

Observation: shopping is finished.

---

## Back Example 2: RFT, reward = 0.75

Task:i am looking for 2 pack of 20ft long quadshield solid copper black color indoor and outdoor coaxial cable, and price lower than 60.00 dollars

Action:Think["response": "I should first use Search to search for 2 pack of 20ft long quadshield solid copper black color indoor and outdoor coaxial cable."]

Observation: OK

Action:search["product": "2 pack of 20ft long quadshield solid copper black color indoor and outdoor coaxial cable"]

Observation: WEB PAGE:
[Back to Search]
Page 1 (Total results: 15)
[Next]
[B074741B7G] PHAT SATELLITE INTL 100ft White Solid Copper COAXIAL RG6 3GHz UV Jacket Indoor Outdoor Satellite TV Antenna Cable 18AWG 75 Ohm Anti-Corrosion Nickel Compression Connectors Assembled in USA $18.0
[B07CQ96G4F] 150ft Black TRI-Shield Indoor Outdoor RG-6 Coaxial Cable Nickel-Plated Brass Connector 75 Ohm (Satellite TV, Broadband Internet, Ham Radio, OTA HD Antenna Coax) Assembled in USA PHAT SATELLITE $19.5
[B07CQB2FZK] 200ft Black TRI-Shield Indoor Outdoor RG-6 Coaxial Cable Nickel-Plated Brass Connector 75 Ohm (Satellite TV, Broadband Internet, Ham Radio, OTA HD Antenna Coax) Assembled in USA PHAT SATELLITE $21.5

Action:Think["response": "None of the products match the exact search criteria. I should click on the first product as it is the closest match."]

Observation: OK

Action:click["button": "B074741B7G"]

Observation: WEB PAGE:
[Back to Search]
[Prev]
size [1ft][1ft (3 pack)][2ft][2ft (3 pack)][3 ft][3ft][3ft (3 pack)][4ft][4ft (3 pack)][5 ft][5ft][5ft (3 pack)][6 ft][6ft][6ft (3 pack)][8ft][8ft (3 pack)][10 ft][10ft][10ft (3 pack)][12 ft][12ft][12ft (3 pack)][15 ft][15ft][15ft (2 pack)][20 ft][20ft][20ft (2 pack)][25 ft][25ft][25ft (2 pack)][30 ft][30ft][30ft (2 pack)][35ft][35ft (2 pack)][40 ft][40ft][40ft (2 pack)][45ft][45ft (2 pack)][50 ft][50ft][50ft (2 pack)][55ft][55ft (2 pack)][60 ft][60ft][60ft (2 pack)][65ft][65ft (2 pack)][70ft][70ft (2 pack)][75ft][75ft (2 pack)][80ft][85ft][90ft][95ft][100 ft][100ft][105ft][110ft][115ft][120ft][125ft][130ft][135ft][140ft][145ft][150 ft][150ft][155ft][160ft][165ft][170ft][175ft][180ft][185ft][190ft][195ft][200 ft][200ft]
color [plenum cmp catvp - white][solid copper - black][solid copper w | weather boot - white][bare copper w | digicon connector - white][direct burial tri-shield - black][direct burial tri-shield, boot - black][quad-shield w | weather boot - black][quadshield - black][quadshield - white][quadshield solid copper - black][solid copper - white]
PHAT SATELLITE INTL 100ft White Solid Copper COAXIAL RG6 3GHz UV Jacket Indoor Outdoor Satellite TV Antenna Cable 18AWG 75 Ohm Anti-Corrosion Nickel Compression Connectors Assembled in USA
Price: $18.0 [Buy Now]

Action:click["button": "20ft (2 pack)"]

Observation: WEB PAGE: You have clicked 20ft (2 pack).

Action:click["button": "solid copper - black"]

Observation: WEB PAGE: You have clicked solid copper - black.

Action:click["button": "Buy Now"]

Observation: shopping is finished.

---

### Back Example 2: EEF, reward = 1.0

Task:i am looking for 2 pack of 20ft long quadshield solid copper black color indoor and outdoor coaxial cable, and price lower than 60.00 dollars

Action:Think["response": "I should use the Search function to find the coaxial cable with the specific requirements."]

Observation: OK

Action:search["product": "20ft long quadshield solid copper black color indoor and outdoor coaxial cable"]

Observation: WEB PAGE:
[Back to Search]
Page 1 (Total results: 15)
[Next]
[B073L44ZBG] 20ft White TRI-Shield Weather Seal Indoor Outdoor RG-6 Coaxial Cable Brass Connector 75 Ohm Satellite TV Broadband Internet Ham Radio OTA HD Antenna Coax Assembled in USA by PHAT SATELLITE INTL $11.4
[B074741B7G] PHAT SATELLITE INTL 100ft White Solid Copper COAXIAL RG6 3GHz UV Jacket Indoor Outdoor Satellite TV Antenna Cable 18AWG 75 Ohm Anti-Corrosion Nickel Compression Connectors Assembled in USA $18.0
[B073L2RDKV] PHAT SATELLITE INTL Tri-Shielded Solid Core 18AWG RG-6 Coaxial Cable, Satellite TV Broadband Modem Internet OTA HD Antenna Indoor Outdoor Coax Applications, UL ETL, Assembled in USA, (200 feet White) $28.5

Action:Think["response": "None of the products match the specific requirements. I should search again with different keywords."]

Observation: OK

Action:search["product": "quadshield solid copper black coaxial cable"]

Observation: WEB PAGE: There is no [Search] button, you should click the [Back to Search] button first to search something

Action:click["button": "Back to Search"]

Observation: WEB PAGE: Let us begin shopping [Search]

Action:search["product": "quadshield solid copper black coaxial cable"]

Observation: WEB PAGE:
[Back to Search]
Page 1 (Total results: 15)
[Next ]
[B07H9G2RPM] PHAT SATELLITE INTL - 3GHz Direct Burial Underground RG6 Coaxial Cable, Tri-Shield Coax, Gel Coated Braids, Weather Boot Compression Connectors, Satellite Approved, Made in USA (12 feet, Orange) $13.0
[B07H9GYZPD] PHAT SATELLITE INTL - 3GHz Direct Burial Underground RG6 Coaxial Cable, Tri-Shield Coax, Gel Coated Braids, Weather Boot Compression Connectors, Satellite Approved, Made in USA (100 feet, Orange) $30.5
[B073KR5D4W] 135ft Black Quad Shield Solid Copper 3GHZ RG-6 Coaxial Cable 75 Ohm (DIRECTV Satellite TV or Broadband Internet) Anti Corrosion Brass Connector RG6 Fittings Assembled in USA by PHAT SATELLITE INTL $52.58

Action:Think["response": "The third product matches the specific requirements. I should click the third product."]

Observation: OK

Action:click["button": "B073KR5D4W"]

Observation: WEB PAGE:
[Back to Search]
[Prev]
size [1ft][2ft][3ft][4ft][5ft][6ft][8ft][10ft][12ft][15ft][20ft][25ft][25 ft][30ft][35ft][40ft][45ft][50ft][55ft][60ft][65ft][70ft][75ft][80ft][85ft][90ft][95ft][100ft][105ft][110ft][115ft][120ft][125ft][130ft][135ft][140ft][145ft][150ft][155ft][160ft][165ft][170ft][175ft][180ft][185ft][190ft][195ft][200ft][210ft][220ft][230ft][240ft][250ft][260ft][270ft][280ft][290ft][300ft][310ft][320ft][330ft][340ft][350ft][360ft][370ft][380ft][390ft][400ft]
style [quadshield - black][quadshield solid copper - black][solid copper core - black][solid copper core - white][solid copper w | weather boot - black][solid copper w | weather boot - white][trishield - black][trishield - white][usa made trishield - black][usa made trishield - white][direct burial - black][direct burial - orange][direct burial 3ghz - orange][direct burial 3ghz w | weather boot - ora...][direct burial rg-11 connector - orange][direct burial rg-11 w | weather boot - or...][direct burial w | rg-11 connector - black][direct burial w | rg-11 weather boot - bl...][direct burial w | weather boot - black][direct burial w | weather boot - orange][plenum rg-11 - white][w | ground - black][w | ground - white][3ghz dual w | ground - black][3ghz w | messenger - black][dual - black][dual copper w | ground - black][dual copper w | ground - white][w | messenger - black] 135ft Black Quad Shield Solid Copper 3GHZ RG-6 Coaxial Cable 75 Ohm (DIRECTV Satellite TV or Broadband Internet) Anti Corrosion Brass Connector RG6 Fittings Assembled in USA by PHAT SATELLITE INTL Price: $52.58
[Buy Now]

Action:click["button": "20ft"]

Observation: WEB PAGE: You have clicked 20ft.

Action:click["button": "quadshield solid copper - black"]

Observation: WEB PAGE: You have clicked quadshield solid copper - black.

Action:click["button": "Buy Now"]

Observation: shopping is finished.

