# OpenReview forum: "Exploring Expert Failures Improves LLM Agent Tuning"
_ICLR.cc/2026/Conference — Submitted to ICLR 2026_

### Official Review · Reviewer_jRTw · 2025-10-24

**Soundness:** 3
**Presentation:** 2
**Contribution:** 3
**Rating:** 6
**Confidence:** 3

**Summary:**

LLM agent training relies on successful expert trajectories. When an expert fails, there is no learning signal, which limits generalisation. However, there may still be useful sub-sequences even within failed expert trajectories, which are currently discarded. The paper proposes a method to reuse these segments to improve learning, identifying beneficial vs harmful actions, and then using selected positive trajectories for fine-tuning. With their new method, they achieve SOTA on a few benchmarks.

**Strengths:**

It's a good idea, learning from failure, or rather extracting value even from failed trajectories. I always wondered why approaches like q-learning which only calculate their score at the very end are so wasteful. This new method is not exactly very complex, but I would count that as an advantage, rather than a disadvantage. The prove is in the results, which seem to confirm the efficacy of their idea, outperforming GPT-4, and showing robustness for different model bases.

**Weaknesses:**

I am not quite sure whether the WebShop and ScienceWorld benchmarks are really enough to show generalisation to open-domain or real-world agent tasks. If the authors truly believe in their paradigm, I would like them to add more benchmarks. Is that feasible as a revision? Would lead me to probably increase my recommendation.

Also, the masking and identification of the beneficial actions are more of a heuristic than a deeper analysis. A bit more reflection would likewise upgrade the paper. This would also help against the possible critique of the missing more thorough hyperparameter sensitivity analysis, especially given the rather small set of benchmarks.

**Questions:**

See weaknesses: Both listed weaknesses could be addressed, questions are written there.

---

> ### Author Response · Authors · 2025-11-21
> **Response**
>
> We sincerely thank the reviewer for the thoughtful and encouraging comments. We especially appreciate your recognition of the importance of extracting useful signals from failed expert trajectories.
>
> In many RL-style methods, rollouts are primarily used to estimate expected returns of current policy (e.g., Q(s,a)), which may be acceptable when in settings with inexpensive simulators, but is wasteful in the LLM regime where every rollout is expensive, especially expert rollouts. At a high level, the core philosophy behind EEF is that an agent’s own blind spots are hard to discover through self-play alone; instead, it should humbly mine every expert attempt for useful ideas, even when those attempts appear to be failures or from a weaker expert.
>
> > C1 I am not quite sure whether the WebShop and ScienceWorld benchmarks are really enough to show generalisation to open-domain or real-world agent tasks.
>
> We fully agree that broader evaluations would strengthen the paper, and we appreciate your openness to reconsidering your recommendation. While extending to additional environments is currently challenging due to the high cost of generating strong-expert trajectories (e.g., ≈$6,000 for WebShop11k) and our limited training resources, we respectfully suggest that WebShop and ScienceWorld already provide meaningful evidence of real-world generalization.
>
> WebShop involves free-form web navigation over ~1M real Amazon products, and WebShop3k explicitly evaluates generalization from a limited set of training tasks to unseen test tasks. WebShop11k further examines scaling and whether the agent becomes a consistently stronger performer.
>
> ScienceWorld is a substantially harder environment than ALFWorld [1] and requires long-horizon (~20 steps) scientific reasoning. Achieving SOTA there without any hyperparameter tuning also suggests robustness rather than environment-specific tuning.
> We will clarify these points in the revision and will actively explore extending EEF to more open-domain and real-world environments in our subsequent work.
>
> [1] AgentGym: Evolving Large Language Model-based Agents across Diverse Environments
>
> > C2 Also, the masking and identification of the beneficial actions are more of a heuristic than a deeper analysis. A bit more reflection would likewise upgrade the paper.
>
> We thank the reviewer for the opportunity to further clarify the masking design, which we agree is essential for making the method more convincing.
>
> Conceptually, EEF treats each s_{\text{need\_recover}} as the beginning of a new, smaller “recovery subtask.” Actions after this state define how to solve the subtask; actions before it are irrelevant for solving this subtask and are therefore masked. This allows the policy to focus on learning how to solve the subtask s_{\text{need\_recover}}.
>
> This design also follows a standard robustness (or generalization) principle in RL: an agent should remain successful even under perturbations (e.g., physically pushing the robot). In our setting, the expert’s behavior plays an analogous role—its actions naturally perturb the trajectory and lead the agent into states it rarely visits. When the agent would normally fail in such a state, we check whether the expert’s next action provides a useful hint for recovery; when it does, we treat that action as beneficial for imitation. By being more robust in a single task, we can expect that it truly learns this task rather than memorizing a single success trajectory, thereby performing better on test performance.

---

> > ### Comment · Reviewer_jRTw · 2025-11-25
> >
> > I appreciate the response and the cost constraints. It is unfortunate that a lot of promising research in our field is de facto gated by requiring access to a lot of resources. I remain confident in my rating and hope the authors can address the concerns of some of my colleagues.

---

> ### Author Response · Authors · 2025-11-26
> **Reponse to Official Comment**
>
> We sincerely appreciate your understanding — in LLM-agent settings, obtaining expert trajectories is indeed extremely costly, typically O(L × N) where L is the total message length and N is the number of turns. This high cost is exactly why we feel that discarding failed expert trajectories is wasteful, and why we developed EEF to recover useful signal from them.
>
> We would also like to briefly add one more insight about our masking strategy. As shown in Figure 4 in appendix, the process can be viewed as a tree search over the failed expert trajectory. The masking ensures that each action in this tree is trained at most once, which helps prevent certain (s_t, a_t) pairs from becoming over-represented and avoids further amplifying simplicity bias.
>
> Since we genuinely believe that simplicity bias and exploration are critical yet under-addressed issues for the community, we will do our best to address the concerns raised by the other reviewers. In particular, we hope to more clearly clarify how our approach differs from prior “avoid-the-error” methods: rather than avoiding replaying the failed segments, EEF selectively reintroduces portions of failed trajectories that still provide useful learning signal, allowing the model to explore beyond the usual easy actions and avoid the reinforcement of simplicity bias that “avoid-the-error” methods can unintentionally cause.
>
> Thank you again for your thoughtful comments.

---

### Official Review · Reviewer_sBXW · 2025-10-29

**Soundness:** 3
**Presentation:** 3
**Contribution:** 3
**Rating:** 6
**Confidence:** 3

**Summary:**

The paper addresses learning from failed expert trajectories when using rejection fine-tuning (RFT). The authors propose EEF (Exploring Expert Failures): run the student from intermediate expert states, identify important states, find trace segments that lead to success when started from those states, mask harmful earlier steps, and add the successful segments to supervised fine-tuning. EEF is simple to add to RFT and yields substantial gains on two agentic benchmarks.

**Strengths:**

- Tackles a practically important problem: exploration in long-horizon, sparse-reward tasks.

- The method is simple and practical to implement on top of existing RFT pipelines.

- The paper includes ablations and diagnostics that help connect the method to the observed performance gains.

- Writing and running examples are clear and help explain the idea.

**Weaknesses:**

Here are some weaknesses that if addressed, can prove EEF’s effectiveness, robustness, and practicality.

- Missing success-rate statistics: The paper does not report how often simulations started from intermediate expert states actually find successful continuations (vs. starting from $s_0$). Without these frequencies it’s unclear whether EEF’s core mechanism is generally effective or only works in selected cases.

- There’s no controlled ablation comparing fine-tuning on $s_0$-only, $s_r$-only, and both. Training only on recovery segments could probably reduces performance from initial states. It could be interesting to understand how much of the performance improvements comes from finetuning on $D_{s_0}$ vs $D_r$.

- Since the paper shows gains using cheaper GPT-3.5 traces, a clearer discussion or an explicit cost breakdown (compute / API dollars / token counts) would help readers weigh spending on higher-quality demonstrations versus more simulation budget. A recommendation for budget allocation under a fixed cost constraint might be especially useful.

- Evaluation covers only two benchmarks; adding another long-horizon domain would help determine whether EEF addresses a general failure mode or a domain-specific phenomenon.

**Questions:**

- Could you provide approximate compute / costs (e.g., API dollars or token counts) for the EEF GPT-4 runs, the RFT×6 baseline, and the mixed GPT-3.5+GPT-4 variant in Table 3?

- Have you observed cases where EEF reduces performance (for example by overfitting to brittle recovery actions)? If so, how common are those cases?

- On “simplicity bias”: is there prior empirical work you can cite? If not, would it be possible to quantify how often failed trajectories in your datasets seem explained by simplicity bias?

- How does RFT perform worse than SFT POS in table 3? Isn’t RFT with one iteration equivalent to SFT POS? Since the paper reports the best model across RFT iterations, RFT should in principle perform at least as well as SFT POS. Could you clarify this discrepancy?

---

> ### Author Response · Authors · 2025-11-21
> **Response 1/2**
>
> We sincerely thank the reviewer for highlighting the importance of long-horizon sparse-reward exploration in multi-task environments and for recognizing that EEF offers a simple and practical way to address the simplicity bias issue we diagnosed.
>
> > C1 Missing success-rate statistics
>
> We thank the reviewer for pointing this out — the success rate on training tasks is indeed crucial, because EEF works by first increasing the number of solvable training subtasks and then relying on the newly learned complex actions to generalize to harder test tasks.
>
> To clarify the importance of starting from intermediate expert states, we will add the statistics from Figure 3, where all simulations are generated using the same SFT agent for a fair comparison. When starting from the initial state, even with 20 rollouts per subtask, only 43% of training tasks yield a positive trajectory, and the resulting RFT model achieves only 38% test success.
>
> In contrast, when simulations start from GPT-3.5 and GPT-4 mid-states, the agent solved 48% and 47% of training tasks respectively; when combined, the success rate rises to 51%. This directly explains why EEF-GPT3, EEF-GPT4, and EEF-GPT3&4 all show substantially stronger test performance. With a better model after each iteration, EEF increasingly solves more previously-unsolved subtasks, creating a positive feedback loop for the next round of training.
>
> We will include these statistics in the revision.
>
> > C2 There’s no controlled ablation comparing fine-tuning on s_0-only, s_r-only, and both
>
> We thank the reviewer for this insightful suggestion—a controlled ablation between s_0-only, s_r-only, and both would indeed be very informative. In our existing experiments, we have already observed that  s_r data is particularly important when leveraging weaker experts (e.g., GPT-3.5) alongside a stronger expert. With s_0-only, EEF (GPT-3.5 & GPT-4) improves over EEF (GPT-4 only) by only about 2%. Once we add s_r trajectories, the improvement increases to around 4–5%. This indicates that s_r trajectories contribute substantially, as weaker experts are more likely to take harmful actions that elicit valuable recovery behaviors from the stronger expert. We will clarify these observations in the revision and, if resources permit, add a more systematic ablation along the lines the reviewer suggested.
>
> > C3 Since the paper shows gains using cheaper GPT-3.5 traces, a clearer discussion or an explicit cost breakdown … A recommendation for budget allocation under a fixed cost constraint might be especially useful.
>
> Thank you for the suggestion. Here’s a detailed cost breakdown:
>
> Using GPT-4-0613 ($30/$60 per 1M input/output tokens) for one trajectory per 11k WebShop tasks cost ~$6,000; GPT-3.5-turbo ($0.50/$1.50 per 1M) for the same was <$200.
>
> For fixed-budget allocation, we recommend a two-phase approach:
>
> **Phase 1 (Strong expert cloning).** Prioritize obtaining at least one high-quality trajectory (e.g., GPT-4) per task and use these demonstrations for behavior cloning to obtain a reasonably strong base agent for subsequent simulations.
>
> **Phase 2 (Diverse exploration).** Then, allocate the remaining budget to cheaper/weaker experts (e.g., GPT-3.5, Gemini Flash) to generate as many additional trajectories as possible, including trajectories that restart from failed strong-expert states, in order to maximize diversity.
>
> We will incorporate this cost breakdown and recommendation into the revised version.
>
> > C4 Evaluation covers only two benchmarks; adding another long-horizon domain would help determine whether EEF addresses a general failure mode or a domain-specific phenomenon.
>
> We sincerely appreciate the reviewer’s suggestion. We fully agree that adding another benchmark would strengthen the evaluation, though suitable environments are currently limited. For example, AgentGym [1] offers 11 environments, but only a few (e.g., WebShop, SciWorld, BIRD) are sufficiently complex, and BIRD is single-turn, making it less suitable for multi-turn agents. Due to this scarcity, we simulate low-data regimes by reducing WebShop tasks to study generalization.
>
> We also note that finetuing multi-turn agent research is resource-intensive, so prior works like ETO and IPR [2] typically evaluate on only a small number of environments (2~3). Nevertheless, we appreciate the suggestion and will aim to incorporate an additional environment in the revision.
>
> [1] AgentGym: Evolving Large Language Model-based Agents across Diverse Environments
>
> [2] Watch Every Step! LLM Agent Learning via Iterative Step-level Process Refinement

---

> ### Author Response · Authors · 2025-11-21
> **Response 2/2**
>
> > Q1 Could you provide approximate compute / costs (e.g., API dollars or token counts) for the EEF GPT-4 runs, the RFT×6 baseline, and the mixed GPT-3.5+GPT-4 variant in Table 3
>
> For the WebShop-11K experiments using 4 NVIDIA A6000 GPUs, the EEF-GPT-4 runs typically completed in about 4 days, the RFT×6 baseline in about 7 days, and the mixed GPT-3.5+GPT-4 variant in about 8 days. The primary reason for this difference is that many of EEF’s simulations start from intermediate expert states, which substantially shortens rollout time.
>
> > Q2 Have you observed cases where EEF reduces performance (for example by overfitting to brittle recovery actions)? If so, how common are those cases?
>
> We rarely observed EEF reducing performance in our current setup. However, in earlier experiments using different solution-selection strategies, we did see instances where solving more training tasks paradoxically hurt test performance. This seemed to occur when the additional solutions were inconsistent, making it harder for the agent to learn them all effectively (cannot solve all solvable training tasks). Based on this observation, we adopted the current approach of selecting solutions closest to the policy distribution, which has effectively mitigated such issues.
>
> > Q3 On “simplicity bias”: is there prior empirical work you can cite?
>
> Yes, prior work includes DART-Math [1], showing rejection-based synthesis biases toward easy queries, and [2], where simpler tasks in multi-task RL cause negative transfer by overshadowing complex ones.
>
> We use a version of RFT from [1] but still see a distribution shift between solvable and normal tasks, indicating persistent simplicity bias.
>
> [1] Dart-math: Difficulty-aware rejection tuning for mathematical problem-solving.
>
> [2] Hard Tasks First: Multi-Task Reinforcement Learning Through Task Scheduling.
>
> We will add them to the revision.
>
> > Q4 How does RFT perform worse than SFT POS in table 3? Isn’t RFT with one iteration equivalent to SFT POS?
>
> Thanks for pointing that out.
> Your understanding is correct: RFT should theoretically perform at least as well as SFT. However, due to retraining from scratch each time, there's normal variance in the reported numbers, leading to slight discrepancies in Table 3. This mirrors Figure 3, where more simulations should improve RFT, but win rates fluctuate due to training stochasticity.

---

### Official Review · Reviewer_VEFG · 2025-10-30

**Soundness:** 2
**Presentation:** 3
**Contribution:** 2
**Rating:** 4
**Confidence:** 4

**Summary:**

The paper introduces Exploring Expert Failures (EEF), a fine-tuning method that enhances the performance of LLM-based agents on complex, multi-step tasks where even expert models (e.g., GPT-4) frequently fail. EEF can expand exploration by reusing expert actions, acknowledging that failed trajectories frequently encode useful signals. The method is evaluated on the WebShop and ScienceWorld benchmarks, demonstrating its effectiveness.

**Strengths:**

The paper is written well and easy to follow.

The idea of reusing valuable actions from failure trajectories to expand exploration sounds promising.

The experiments on the WebShop and ScienceWorld benchmarks demonstrate the effectiveness of the method.

**Weaknesses:**

The novelty of the proposed method appears limited, as the key concept of utilizing failure trajectories is not new. It should be noted that similar ideas have been investigated in prior works, including IPR [1], LEMA [2], and STeCa [3].

The experiments are not sufficiently comprehensive. (1) Several related works are not compared against, such as IPR [1], LEMA [2], and STeCa [3]. (2) Although LLAMA3-8B Instruct and Mistral-7B-v0.3 were used as base models, the results for Mistral-7B-v0.3 are not presented in sufficient detail. Moreover, since Mistral-7B-v0.3 was released a year ago, it is recommended to conduct experiments using Qwen3-8B as an additional base model. (3) In the experiments, a fair comparison should be made (with Iter=3) against the method that only training on solutions from the initial state. (4) It is necessary to compare EEF with different solution selection strategies, such as the randomly selecting strategy. (5) The individual effects of the two types of important states, $D_{s_0}$ and $D_r$, should be verified to understand their respective impacts on the experiment. (6) The analysis experiments currently focus only on WebShop. Similar analyses should be extended to the ScienceWorld benchmark. (7) In the Efficiency Analysis, a brief discussion on the trade-off between the simulation budget (M) and performance gains would be helpful.

Regarding the methodology, harmful states are identified through agent simulation. Could sampling issues potentially lead to inaccurate judgments in this process?

[1] Watch Every Step! LLM Agent Learning via Iterative Step-level Process Refinement, EMNLP 2024 \
[2] Learning From Mistakes Makes LLM Better Reasoner, 2023  \
[3] STeCa: Step-level Trajectory Calibration for LLM Agent Learning, 2025.2

**Questions:**

See the weaknesses.

---

> ### Author Response · Authors · 2025-11-21
> **Response 1/2**
>
> We thank the reviewer for the insightful comments. Many of the concerns stem from assumptions about the goals and mechanisms of EEF that differ from those in prior negative-sample or stepwise methods. We clarify these distinctions below and address all questions in a point-by-point manner.
>
> > C1 The novelty of the proposed method appears limited …  similar ideas have been investigated in prior works, including IPR [1], LEMA [2], and STeCa [3].
>
> We thank the reviewer for the concern but believe EEF is conceptually and mechanistically distinct from prior stepwise/negative-sample methods:
>
> 1. **Fundamentally different objectives**. Approaches like IPR [1], LEMA [2], STeCa [3], ETO, and NAT leverage failed trajectories primarily to teach the model to avoid erroneous actions. In contrast, EEF is specifically designed to combat simplicity bias—where behaviors learned from simple tasks negatively transfer to harder ones—arising from the lack of positive expert demonstrations in challenging tasks. This error-avoidance objective does not address the bias and, as we empirically demonstrate for ETO and NAT, can actually exacerbate it: their models adopt navigation actions even less frequently than standard RFT on hard tasks (see Table 4).
>
> 2. **Opposite effect on complex actions**, even with stepwise analysis. Because the goal of stepwise methods (e.g., IPR, STeCa) remains error avoidance, they tend to identify navigation actions as the “cause of failure” in simple tasks where such actions are indeed unnecessary and harmful. This correctly penalizes navigation in easy tasks but also causes the model to systematically under-explore these actions in harder tasks that actually require them. EEF takes the opposite approach: it deliberately re-evaluates and resurrects seemingly erroneous actions from past failures in later iterations once the agent has gained the capability to execute them successfully.
>
> 3. **Retention of expert failed trajectories.** Unlike ETO, IPR, and STeCa, which discard **expert** failures entirely and rely only on self-generated failures, EEF purposefully retains and re-uses them. These expert failures are precious sources of rare complex actions that never appear in successful expert trajectories and thus receive no positive supervision in their datasets.
>
> In summary, prior methods use failed trajectories mainly to learn “what not to do”, while EEF uses them to discover “what was prematurely judged as wrong but will become correct later”. This reversal of the learning objective—together with our iteration-aware re-evaluation mechanism—makes EEF fundamentally distinct from existing negative-sample and stepwise approaches. Additional discussion is provided in the related-work section (lines 463–467), which explains why these stepwise methods are not able to leverage failed expert trajectories in the manner EEF does.
>
> > Q1 Several related works are not compared against, such as IPR [1], LEMA [2], and STeCa [3].
>
> We thank the reviewer for the suggestion. As discussed in our response to C1, EEF targets a problem that is largely orthogonal to IPR, LEMA, and STeCa: mitigating **simplicity bias** in challenging LLM-agent environments where expert demonstrations frequently fail on OOD/hard tasks and provide no positive rewards for complex actions (e.g., navigation). In contrast, many of the environments used in IPR/LEMA/STeCa feature much stronger expert-demo coverage of the target distribution (e.g., human demonstrations in AlfWorld, or math benchmarks where strong LLMs already achieve high accuracy), making direct comparisons in our expert-failure-heavy setting less informative.
>
> Crucially, these methods can in fact benefit from EEF: our method first generates positive demonstrations for missing behaviors in such challenging environments, thereby enriching the data on which IPR/LEMA/STeCa can subsequently be applied.
>
> > Q2 Mistral-7B-v0.3 are not presented in sufficient detail. It is recommended to conduct experiments using Qwen3-8B as an additional base model.
>
> We thank the reviewer for the suggestion. We will clarify the details of the Mistral‑7B setup in the revision and, if resources permit, add results on a stronger Qwen3-8B model.
> Our current choice of Llama‑2/3 and Mistral‑7B‑v0.3 is deliberate: (i) these backbones are shared with recent methods like IPR and STeCa, enabling fair, apples‑to‑apples comparison with prior work; and (ii) they show that EEF can substantially improve relatively weak models on challenging problems, isolating the effect of our algorithm rather than relying on the capabilities of the latest LLM.

---

> ### Author Response · Authors · 2025-11-21
> **Response 2/2**
>
> > Q3 In the experiments, a fair comparison should be made (with Iter=3) against the method that only training on solutions from the initial state.
>
> We thank the reviewer for this suggestion. Since EEF is built on top of RFT, we have ensured that the RFT-based baselines were trained with sufficient computational budget (version RFTx6). Empirically, allocating more resources to these methods does not yield improvements; in fact, for both RFT and ETO, increasing the number of iterations or sampling more trajectories per iteration sometimes actually degrades test performance. For instance, in the original ETO paper, the method performs best at Iter=2 rather than Iter=3. This might be due to overfitting the training tasks.  We will clarify this behavior in the revised manuscript and include a brief ablation study to document these trends.
>
> > Q4 It is necessary to compare EEF with different solution selection strategies, such as the randomly selecting strategy.
>
> We appreciate the reviewer’s thoughtful suggestion and agree that exploring alternative selection strategies could potentially yield even better performance. Our focus in this work, however, is to demonstrate that simplicity bias is a fundamental issue in existing methods, and that explicitly addressing it leads to clear improvements. While different selection strategies may further enhance results, they do not affect the validity of our core claim.
>
> > Q5 The individual effects of the two types of important states,  and should be verified to understand their respective impacts on the experiment.
>
> We thank the reviewer for the suggestion. We will include additional ablations in the revision. Our internal results show that s_{need _recovery} yields minimal gains on pure GPT-4 datasets, but provides clear improvements on mixed GPT-3 + GPT-4 settings.
>
> > Q6 The analysis experiments currently focus only on WebShop. Similar analyses should be extended to the ScienceWorld benchmark.
>
> We thank the reviewer for the suggestion. Preliminary analyses are provided in Appendix A.7. Unlike WebShop, ScienceWorld lacks a dominant reusable complex action, making statistical analysis less straightforward, though we still observe signs of simplicity bias.
>
> >  Q7 In the Efficiency Analysis, a brief discussion on the trade-off between the simulation budget (M) and performance gains would be helpful. Regarding the methodology, harmful states are identified through agent simulation. Could sampling issues potentially lead to inaccurate judgments in this process?
>
> We thank the reviewer for this suggestion. Regarding the trade-off, when the simulation budget (M) is small, there may be many actions between consecutive simulations, potentially leading the agent to misjudge an entire segment as beneficial when it actually contains harmful states (or vice versa). Therefore, simulating from different sets of expert states​ in every iteration could be a better approach. However, this does not affect our main claim.

---

### Official Review · Reviewer_m5Mf · 2025-11-01

**Soundness:** 2
**Presentation:** 2
**Contribution:** 2
**Rating:** 2
**Confidence:** 3

**Summary:**

The paper introduces a method called Exploring Expert Failures (EEF) to improve the fine-tuning of LLM agents. The authors address a key limitation of standard techniques like Rejection Sampling Fine-Tuning (RFT), where agents learn only from successful expert demonstrations and thus fail to master complex tasks where the expert often fails. The core idea of EEF is to salvage useful information from the expert's failed trajectories by simulating the agent's performance from intermediate steps. By identifying segments of a failed path that can lead to success, EEF incorporates these "beneficial actions" into the training data, demonstrably improving performance on benchmarks like WebShop and SciWorld.

**Strengths:**

The problem studied in the paper is very interesting, especially the analysis of RFT, although it seems that this problem has not been well solved.

**Weaknesses:**

The methodology's innovation is arguably incremental. It builds directly upon the existing RFT paradigm, and its primary contribution is a more sophisticated data filtering and augmentation strategy rather than a fundamentally new approach to agent learning. The effectiveness of EEF is heavily dependent on extensive simulation and sampling. The process requires re-simulating trajectories from numerous states within failed expert attempts to identify useful sub-paths, raising concerns about its computational expense and scalability. The performance gains are achieved through what is essentially a more guided, brute-force exploration of the expert's failure space. In essence, the method refines the "guess-and-check" nature of sampling-based tuning by adding a more targeted "check" phase, but it does not move beyond this data-intensive framework.


1. What if the model consistently fails to sample a successful trajectory?

2. Is there an analysis of the sampling efficiency? Specifically, what proportion of the explored trajectories are ultimately found to be useful for training?

3. As I mentioned in the summary, this method relies on extensive sampling and is confined to Supervised Fine-Tuning.

4. From another perspective, the method like GRPO also samples a large number of rollouts but then uses Reinforcement Learning to optimize the model, rather than SFT. This appears to be a more logical approach.

**Questions:**

Stated in Weaknesses

---

> ### Author Response · Authors · 2025-11-21
> **Response 1/2**
>
> We thank the reviewer for your detailed feedback and insightful questions. We appreciate the opportunity to clarify our contributions and address the concerns raised.
>
> > C1 The methodology's innovation is arguably incremental. It builds directly upon the existing RFT paradigm
>
> While EEF builds on RFT, its core innovation addresses a critical gap in the no-success regime for challenging tasks, where pure self-play rarely discovers rewards and instead amplifies **simplicity bias**. This issue affects many methods, including RL-based approaches. For example, in Figure 3, even with 20 simulations per training subtask, 59% of tasks remain unsolved (i.e., 0% success rate and no expert success trajectories), causing the agent to reinforce easy behaviors while potentially eroding actions required for harder tasks (e.g., navigation actions in Table 4). In this regime, there is no mechanism—even in RL methods—to bootstrap learning on the unsolved tasks.
>
> EEF tackles this simplicity bias issue orthogonally in two ways:
>
> (1) Exploration from expert failures: By reusing failed expert trajectories as seeds (starting from intermediate ChatGPT-3.5/4 states), we solve ~15% more training tasks with just 5 simulations per subtask versus 20× self-simulation from scratch, converting unreachable hard subtasks into reward-bearing data.
>
> (2) Recovery learning: For "solved" subtasks, agents often memorize single paths and fail on out-of-distribution states; EEF identifies recovery states in expert trajectories where the agent failed and teaches the agent with expert’s "recovery" behaviors.
>
> These mechanisms enable training signals for previously unsolved subtasks and states, essential for any environment interactive agent (including GRPO/ETO), improving generalization to unseen test tasks. EEF is thus a general recipe for hard-task bootstrapping, not merely trajectory filtering or augmentation—we use RFT as a simple baseline to showcase it, but it extends beyond.
>
> > C2 The effectiveness of EEF is heavily dependent on extensive simulation and sampling.
>
> We appreciate the reviewer's concern but respectfully disagree that EEF "heavily depends on extensive simulation and sampling." EEF is more sample-efficient than pure self-play methods: Figure 3 shows that with just 2 simulations per task (M=1: one self-play, one from mid-expert failure), EEF outperforms an RFT baseline needing 20 self-play rollouts.
>
> Here we want to clarify that sample efficiency can be viewed in two ways:
> (1) the fraction of samples that are actually utilized in training, and
> (2) the performance improvement achieved per sample generated.
>
> Even if a method uses all generated samples, poorly structured samples (e.g., unguided self-play) may provide limited learning signal. In contrast, EEF leverages informative simulations seeded from expert failures, producing samples that more consistently lead to meaningful updates and thus requiring fewer simulations overall.
>
> In addition, common RL methods such as GRPO typically require on the order of 16–50 simulations per task to estimate preference scores and gradients, and need more iterations compared to EEF (iter=3). EEF's edge comes from targeted exploration via expert failures, not extensive simulation.
>
> > C3 In essence, the method refines the "guess-and-check" ... it does not move beyond this data-intensive framework.
>
> Although we carefully identify learnable trajectory segments, our main contribution is tackling simplicity bias in the no-success regime (discussed in C1).
>
> >  Q1 What if the model consistently fails to sample a successful trajectory?
>
> We thank the reviewer for this insightful question. If the model never samples a successful trajectory, EEF cannot generate signal from nothing, as it relies on some beneficial actions in expert data; in such cases, more (or diverse) expert rollouts would be needed. Fortunately, our experiments demonstrate EEF's robustness: even a weak expert like GPT-3.5 (23% win rate) provides useful signals (e.g., in ablations showing no performance loss vs. stronger experts), and combining multiple experts of varying strengths further boosts results.

---

> ### Author Response · Authors · 2025-11-21
> **Response 2/2**
>
> >  Q2   Is there an analysis of the sampling efficiency? Specifically, what proportion of the explored trajectories are ultimately found to be useful for training?
>
> Although sampling efficiency (first type of sampling efficiency in C2) is important, in multi-task settings naively maximizing the proportion of trajectories used for training can easily induce simplicity bias. If we were to use all positive trajectories, the positive training signal would tend to be dominated by simple tasks, since they are easier to solve and therefore overrepresented among successes [1]. Conversely, if we were to retain all negative trajectories, the model could become overly conservative in ways that suppress the complex action sequences required for harder tasks (e.g., navigation actions). To balance exploration and robustness, EEF therefore discards all failed trajectories and additionally sub-samples successful ones when constructing the training set. For reference, in the main EEF GPT‑4 experiment, only about 15% of the generated trajectories are used for training in the first iteration. In future work, we plan to explore incorporating RL-style techniques to improve sampling efficiency while explicitly mitigating simplicity bias, in the spirit of difficulty-aware approaches such as Dart-math [1].
>
> [1] Dart-math: Difficulty-aware rejection tuning for mathematical problem-solving.
>
> > Q3   As I mentioned in the summary, this method relies on extensive sampling and is confined to Supervised Fine-Tuning.
>
> As discussed in C2, EEF does not rely on extensive sampling; it is in fact more sample-efficient than baselines and other RL methods. Additionally, as noted in C1, we use RFT merely as a simple baseline to demonstrate EEF's core concepts, which extend beyond it. Finally, echoing reviewer jRTw's point, we do not view the simplicity of methods like SFT/RFT as a drawback; on the contrary, it offers key advantages in many scenarios, enabling faster and more stable deployment.
>
> > Q4 The method like GRPO also samples a large number of rollouts but then uses Reinforcement Learning to optimize the model, rather than SFT. This appears to be a more logical approach.
>
> We view EEF as complementary to RL: the positive trajectories discovered by EEF from expert failures can serve as high-value rollouts for methods like GRPO in future work. In addition, pure GRPO is not well suited to our problem setting because: (1) it can suffer from instability due to entropy collapse in sparse-reward environments; (2) typical GRPO deployments require substantially more simulations than RFT/ETO/EEF-style methods; and (3) iterative GRPO training tends to produce very long reasoning chains, which might result in GPU memory issue.

---

### Meta-Review · Area_Chair_q4hA · 2026-01-03

**Summary:**

This paper proposes Exploring Expert Failures (EEF), a fine-tuning approach for LLM-based agents in long-horizon, sparse-reward environments. The key idea is to mine useful learning signals from failed expert trajectories by restarting the agent from intermediate expert states, identifying recoverable segments that can lead to success, masking earlier steps, and incorporating these recovered segments into supervised fine-tuning.

Reviewers generally agree that the problem addressed is important and that learning from failed expert trajectories is a natural and practically motivated direction. The method is simple, easy to integrate into existing pipelines, and empirically effective on the tested benchmarks. However, despite thoughtful and detailed author responses, several substantive concerns remain that prevent recommending acceptance at ICLR.

1. Limited Novelty Relative to Prior Work.
   Although the authors argue that EEF is conceptually distinct from prior failure-aware or step-level methods (e.g., IPR, LEMA, STeCa, ETO), the methodological gap remains insufficiently convincing. At a high level, EEF still operates within the same paradigm of sampling trajectories, filtering or reweighting them, and applying supervised fine-tuning. The claimed distinction—addressing “simplicity bias” rather than error avoidance—relies largely on qualitative argumentation. Without direct, controlled comparisons to the most closely related methods, it remains unclear whether the gains stem from a fundamentally new insight or from more aggressive and targeted data reuse.

2. Heuristic Design and Lack of Theoretical Grounding.
   The identification of “beneficial” versus “harmful” states and the masking strategy are largely heuristic. While intuitively reasonable, the paper does not provide a principled justification or theoretical analysis explaining why these heuristics should reliably isolate useful learning signals. As a result, it is difficult to assess the robustness of the approach or to generalize it beyond the specific benchmarks studied.

3. Confinement to Supervised Fine-Tuning and Scalability Concerns.
   The approach remains restricted to supervised fine-tuning and depends on repeated simulation from expert trajectories. While the authors argue that EEF is more sample-efficient than unguided self-play or some RL methods, concerns remain about scalability to larger environments or more complex tasks, particularly when expert failures are frequent and successful continuations are rare. The relationship between EEF and reinforcement-learning-based solutions is left largely conceptual rather than empirically grounded.

4. Insufficient Breadth of Empirical Validation.
   The evaluation is limited to two benchmarks. Although both are widely used, this scope is not sufficient to support broader claims about addressing a general failure mode of LLM-agent training. Several requested ablations and comparisons—such as isolating the effects of different recovered trajectory types, evaluating robustness across additional domains, or comparing against closely related baselines—remain incomplete or absent.

In summary, while this work presents a reasonable and practically useful extension to existing fine-tuning practices, the remaining concerns regarding novelty, heuristic methodology, limited empirical breadth, and unclear generality prevent a positive recommendation. The paper would benefit from stronger differentiation from prior work, broader and more rigorous empirical validation, and clearer principled grounding. For these reasons, I recommend **rejection** at this time, with encouragement to resubmit after addressing the above issues.

**Reviewer Concerns:**

As above

**Reviewer Scores:**

I do think the reviewers would change their scores.

---

### Decision · Program_Chairs · 2026-01-26

Reject